# ORTHOREG: IMPROVING GRAPH-REGULARIZED MLPS VIA ORTHOGONALITY REGULARIZATION

## ABSTRACT

Graph Neural Networks (GNNs) are currently dominating in modeling graph-structure data, while their high reliance on graph structure for inference significantly impedes them from widespread applications. By contrast, Graph-regularized MLPs (GR-MLPs) implicitly inject the graph structure information into model weights, while their performance can hardly match that of GNNs in most tasks. This motivates us to study the causes of the limited performance of GR-MLPs. In this paper, we first demonstrate that node embeddings learned from conventional GR-MLPs suffer from dimensional collapse, a phenomenon in which the largest a few eigenvalues dominate the embedding space, when a linear encoder is used. As a result of this the expressive power of the learned node representations is constrained. We further propose ORTHO-REG, a novel GR-MLP model, to mitigate the dimensional collapse issue. Through a soft regularization loss on the correlation matrix of node embeddings, ORTHO-REG explicitly encourages orthogonal node representations and thus can naturally avoid dimensionally collapsed representations. Experiments on traditional transductive semi-supervised classification tasks and inductive node classification for cold-start scenarios demonstrate its effectiveness and superiority.

## 1 INTRODUCTION

Graph Machine Learning (GML) has been attracting increasing attention due to its wide applications in many real-world scenarios, like social network analysis (Fan et al., 2019), recommender systems (van den Berg et al., 2017; Wu et al., 2019b), chemical molecules (Wang et al., 2021; Stärk et al., 2022) and biology structures. Graph Neural Networks (GNNs) (Kipf & Welling, 2017; Hamilton et al., 2017; Velickovic et al., 2018; Xu et al., 2019) are currently the dominant models for GML thanks to their powerful representation capability through iteratively aggregating information from neighbors. Despite their successes, such an explicit utilization of graph structure information hinders GNNs from being widely applied in industry-level tasks. On the one hand, GNNs rely on layer-wise message passing to aggregate features from the neighborhood, which is computationally inefficient during inference, especially when the model becomes deep (Zhang et al., 2021). On the other hand, recent studies have shown that GNN models can not perform satisfactorily in cold-start scenarios where the connections of new incoming nodes are few or unknown (Zheng et al., 2021). By contrast, Multi-Layer Perceptrons (MLPs) involve no dependence between pairs of nodes, indicating that they can infer much faster than GNNs (Zhang et al., 2021). Besides, they can predict for all nodes fairly regardless of the numbers of connections, thus can infer more reasonably when neighborhoods are missing (Zheng et al., 2021). However, it remains challenging to inject the knowledge of graph structure information into learning MLPs.

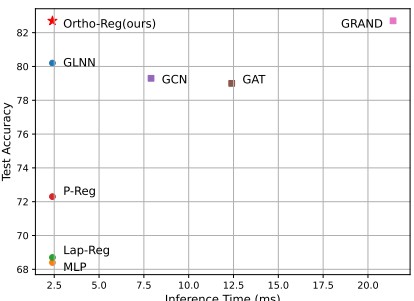

Figure 1: As an MLP model, our method performs even better than GNN models on `Pubmed`, but with a much faster inference speed. GRAND (Feng et al., 2020) is one of the SOTA GNN models on task. Circled markers denote MLP baselines, and squared markers indicate GNN baselines.

One classical and popular method to mitigate this issue is Graph-Regularized MLPs (GR-MLPs in short). Generally, besides the basic supervised loss (e.g., cross-entropy), GR-MLPs employ

an additional regularization term on the final node embeddings or predictions based on the graph structure (Ando & Zhang, 2006; Zhou et al., 2003; Yang et al., 2021; Hu et al., 2021). Though having different formulations, the basic idea is to make node embeddings/predictions smoothed over the graph structure. Even though these GR-MLP models can implicitly encode the graph structure information into model parameters, there is still a considerable gap between their performance compared with GNNs (Ando & Zhang, 2006; Yang et al., 2021). Recently, another line of work, GNN-to-MLP knowledge distillation methods (termed by KD-MLPs) (Zhang et al., 2021; Zheng et al., 2021), have been explored to incorporate graph structure with MLPs. In KD-MLPs, a student MLP model is trained using supervised loss and a knowledge-distillation loss from a well-trained teacher GNN model. Empirical results demonstrate that with merely node features as input, the performance of KD-MLPs can still match that of GNNs as long as they are appropriately learned. However, the 2-step training of KD-MLPs is undesirable, and they still require a well-trained GNN model as a teacher. This motivates us to rethink the failure of previous GR-MLPs to solve graph-related applications and study the reasons that limit their performance.

**Presented work:** In this paper, we first demonstrate that node embeddings learned from existing GR-MLPs suffer from dimensional collapse (Hua et al., 2021; Jing et al., 2022), a phenomenon that the embedding space of nodes is dominated by the largest (a few) eigenvalue(s). Our theoretical analysis demonstrates that the dimensional collapse in GR-MLP is due to the irregular feature interaction caused by the graph Laplacian matrix (see Lemma 1). We then propose Orthogonality Regularization (ORTHO-REG in short), a novel GR-MLP model, to mitigate the dimensional collapse issue in semi-supervised node representation learning tasks. The key design of ORTHO-REG is to enforce an additional regularization term on the output node embeddings, making them **orthogonal** so that different embedding dimensions can learn to express various aspects of information. Besides, ORTHO-REG extends the traditional first-order proximity preserving target to a more flexible one, improving the model's expressive power and generalization ability to non-homophily graphs. We provide a thorough evaluation for ORTHO-REG on various node classification tasks. The empirical results demonstrate that ORTHO-REG can achieve competitive or even better performance than GNNs. Besides, using merely node features to make predictions, ORTHO-REG can infer much faster on large-scale graphs and make predictions more reasonable for new nodes without connections. In Fig. 1 we present the performance of ORTHO-REG compared with GNNs and other MLPs on `Pubmed`, where ORTHO-REG achieves SOTA performance with the fastest inference speed.

**We summarize our contributions as follows**:

1) We are the first to examine the limited representation power of existing GR-MLP models from the perspective of dimensional collapse. We provide theoretical analysis and empirical studies to justify our claims.

2) To mitigate the dimensional collapse problem, we design a novel GR-MLP model named ORTHO-REG. ORTHO-REG encourages the node embeddings to be orthogonal through explicit soft regularization, thus can naturally avoid dimensional collapse.

3) We conduct experiments on traditional transductive semi-supervised node classification tasks and inductive node classification under cold-start scenarios on public datasets of various scales. The numerical results and analysis demonstrate that by learning orthogonal node representations, ORTHO-REG can outperform GNN models on these tasks.

## 2 BACKGROUNDS AND RELATED WORKS

### 2.1 PROBLEM FORMULATION

We mainly study a general semi-supervised node classification task on a single homogeneous graph where we only have one type of node and edge. We denote a graph by $\mathcal{G} = (\mathcal{V}, \mathcal{E})$, where $\mathcal{V}$ is the node set, and $\mathcal{E}$ is the edge set. For a graph with $N$ nodes (i.e., $|\mathcal{V}| = N$), we denote the node feature matrix by $\boldsymbol{X} \in \mathbb{R}^{N \times D}$, the adjacency matrix by $\boldsymbol{A} \in \mathbb{R}^{N \times N}$. In semi-supervised node classification tasks, only a small portion of nodes are labeled, and the task is to infer the labels of unlabeled nodes using the node features and the graph structure. Denote the labeled node set by $\mathcal{V}^L$ and the unlabeled node set by $\mathcal{V}^U$, then we have $\mathcal{V}^L \cap \mathcal{V}^U = \varnothing$ and $\mathcal{V}^L \cup \mathcal{V}^U = \mathcal{V}$.

Denote the one-hot ground-truth labels of nodes by $\hat{\boldsymbol{Y}} \in \mathbb{R}^{N \times C}$, and the predicted labels by $\boldsymbol{Y}$. One can learn node embeddings $\boldsymbol{H}$ using node features $\boldsymbol{X}$ and adjacency matrix $\boldsymbol{A}$, and use the

embeddings to generate predicted labels $\hat{\boldsymbol{Y}}$. For example, GNNs generate node representations through iteratively aggregating and transforming the embeddings from the neighbors and could be generally formulated as $\boldsymbol{H} = f_\theta(\boldsymbol{X}, \boldsymbol{A})$. Then a linear layer is employed on top of node embeddings to predict the labels $\boldsymbol{Y} = g_\theta(\boldsymbol{H})$. The model could be trained in an end-to-end manner by optimizing the cross-entropy loss between predicted labels and ground-truth labels of labeled nodes: $\mathcal{L}_{sup} = \ell_{xent}(\boldsymbol{Y}^L, \hat{\boldsymbol{Y}}^L) = \sum_{i \in \mathcal{V}^L} \ell_{xent}(\boldsymbol{y}_i, \hat{\boldsymbol{y}}_i)$. Note that GNNs explicitly utilize the graph structure information through learning the mapping from node features and graph adjacency matrix to predicted labels. However, due to the limitations introduced in Sec. 1 (inefficiency at inference and poor performance for cold-start nodes), we seek to learn an MLP encoder, i.e., $\boldsymbol{H} = f_\theta(\boldsymbol{X})$ that only takes node features for making predictions.

## 2.2 GRAPH-REGULARIZED MLPs

Graph-Regularized MLPs (GR-MLPs in short) implicitly inject the graph knowledge to the MLP model with an auxiliary regularization term on the node embeddings/predictions over the graph structure (Zhou et al., 2003; Yang et al., 2021; Hu et al., 2021), whose objective function could be generally formulated as: $\mathcal{L} = \mathcal{L}_{sup} + \lambda \mathcal{L}_{reg}$, where $\mathcal{L}_{reg} = \ell(\boldsymbol{H}, \boldsymbol{A})$ or $\ell(\boldsymbol{Y}, \boldsymbol{A})$. The most representative graph regularization method, Graph Laplacian Regularization (Zhou et al., 2003; Ando & Zhang, 2006), enforces local smoothness of embeddings/predictions between two connected nodes: $\ell(\boldsymbol{Y}, \boldsymbol{A}) = \text{tr}[\boldsymbol{Y}^\top \boldsymbol{L} \boldsymbol{Y}]$, where $\boldsymbol{L} = \boldsymbol{I} - \tilde{\boldsymbol{A}} = \boldsymbol{I} - \boldsymbol{D}^{-1/2} \boldsymbol{A} \boldsymbol{D}^{-1/2}$ is the (symmetric normalized) Laplacian matrix of the graph. Note that $\boldsymbol{Y}$ can be replaced with $\boldsymbol{H}$ if one would like to regularize node embeddings instead of predicted labels.

Later works apply advanced forms of regularization, like propagation regularization (P-Reg, Yang et al. (2021)), contrastive regularization (Hu et al., 2021), etc. Regardless of the minor differences, they are all based on the graph homophily assumption that connected nodes should have similar representations/labels. With the graph structure information implicitly encoded into the model parameters, GR-MLPs can improve the representative power of MLP encoders. However, their performances are still hard to match compared to those of GNN models.

**Remark 1.** (Differences from Graph-Augmented MLPs). *Though sound similar, Graph-regularized MLPs(GR-MLPs) are totally different from Graph-augmented MLPs (GA-MLPs). Although trained with implicit graph structure regularization, GR-MLPs make predictions directly through the MLP model. By contrast, GA-MLPs, such as SGC (Wu et al., 2019a), APPNP (Klicpera et al., 2019), GFNN (NT & Maehara, 2019) and SIGN (Rossi et al., 2020) explicitly employs the graph structure to augment the node representation generated from an MLP model.*

## 2.3 DIMENSIONAL COLLAPSE

Dimensional collapse (also known as spectral collapse in some work (Liu et al., 2019)) is a phenomenon in representation learning where the embedding space is dominated by the largest a few singular values (other singular values decay significantly as the training step increases). As the actual embedding dimension is usually large, the dimensional collapse phenomenon prevents different dimensions from learning diverse information, limiting their representation power and ability to be linearly discriminated. Jing et al. (2022) has analyzed the dimensional collapse phenomenon from a theoretical perspective and attributed it to the effect of strong data augmentation and implicit regularization effect of neural networks (Arora et al., 2019; Ji & Telgarsky, 2019). Previous methods usually adopt whitening operation (Hua et al., 2021; Ermolov et al., 2021) to mitigate this issue, while such explicit whitening methods are usually computationally inefficient and thus are not applicable to GR-MLP where efficiency is much more important. In this paper, we demonstrate that node embeddings learned from conventional Graph-Regularized MLPs also suffer from dimensional collapse. We provide a theoretical analysis on how it is caused and develop a computationally efficient soft regularization term to mitigate it.

## 3 ISSUE DETECTION: DIMENSIONAL COLLAPSE IN GR-MLPS

In this section, we investigate the reasons behind the weak representation power of previous GR-MLPs. In short, we find that node embeddings learned through traditional GR-MLPs (e.g., with graph Laplacian regularization (Ando & Zhang, 2006)) suffer from dimensional collapse, a phenomenon

which is first investigated in contrastive self-supervised learning (He & Ozay, 2022; Hua et al., 2021; Jing et al., 2022). Generally speaking, dimensional collapse indicates that the embedding space is dominated by the largest few eigenvalues, limiting their representation powers. We then show that the dimensional collapse phenomenon does exist in the typical GR-MLP model, Graph Laplacian Regularization (Ando & Zhang, 2006), from both theoretical analysis and empirical justification. We provide further discussion on the impacts of dimensional collapse on downstream classification tasks in Appendix E. The objective function of Graph Laplacian Regularization for semi-supervised node classification tasks could be formulated as follows:

$$\mathcal{L} = \ell_{xent}(\boldsymbol{Y}^L, \hat{\boldsymbol{Y}}^L) + \lambda \text{tr}[\boldsymbol{H}^\top \boldsymbol{L}\boldsymbol{H}] = \sum_{i \in \mathcal{V}^L} \ell_{xent}(\boldsymbol{y}_i, \hat{\boldsymbol{y}}_i) + \lambda \sum_{i,j \in \mathcal{V}} A_{ij} \| \frac{\boldsymbol{h}_i}{\sqrt{D_{ii}}} - \frac{\boldsymbol{h}_j}{\sqrt{D_{jj}}} \|^2. \quad (1)$$

We would like to study the effect of the Laplacian regularization term $\mathcal{L}_{reg} = \text{tr}[\boldsymbol{H}^\top \boldsymbol{L}\boldsymbol{H}]$ on $\boldsymbol{H}$'s embedding space through analyzing the eigenspectra of its auto-correlation matrix $\boldsymbol{C} = \{C_{kk'}\}, \in \mathbb{R}^{d \times d}$, where $c_{kk'}$ is defined as:

$$C_{kk'} = \frac{\Sigma_{kk'}}{\sqrt{\Sigma_{kk}\Sigma_{k'k'}}}, \text{ and } \boldsymbol{\Sigma} = \sum_{i \in \mathcal{V}} \frac{(\boldsymbol{h}_i - \overline{\boldsymbol{h}})(\boldsymbol{h}_i - \overline{\boldsymbol{h}})^\top}{|\mathcal{V}|} \quad (2)$$

Note that $\overline{\boldsymbol{h}} = \sum_{i=1}^{|\mathcal{V}|} \boldsymbol{h}_i / |\mathcal{V}|$ is the average node embedding vector, so $\boldsymbol{\Sigma}$ is the covariance matrix of $\boldsymbol{H}$, and we denote $\boldsymbol{C}$'s eigenvalues in a descending order by $\{\lambda_1^C, \lambda_2^C, \cdots, \lambda_D^C\}$.

**Theoretical Analysis.** To simplify our analysis, we consider a simple single-layer Perceptron (linear) model as the encoder to learn node embeddings, i.e., $\boldsymbol{H} = \boldsymbol{X}\boldsymbol{W}$ (note that we validate the non-linear case empirically in the empirical justification part below), where $\boldsymbol{W} \in \mathbb{R}^{F \times D}$ is the weight matrix (we further assume $F = D$ in this part for simplicity). The model (i.e., the weight matrix $\boldsymbol{W}$) is optimized using stochastic gradient descent. Then we have the following lemma on the evolvement of the weight matrix's singular values.

**Lemma 1.** (Shrinking singular-space of weight matrix.) *Consider the linear model above which is optimized with $\mathcal{L}_{reg} = tr[\boldsymbol{H}^\top \boldsymbol{L}\boldsymbol{H}]$. Let $\boldsymbol{P} = \boldsymbol{X}^\top \boldsymbol{L}\boldsymbol{X} = \sum_{ij} L_{ij}\boldsymbol{x}_i \cdot \boldsymbol{x}_j^\top$ and denote its non-ascending eigenvalues by $\{\lambda_1^P, \lambda_2^P, \cdots, \lambda_D^P\}$. Denote the randomly initialized weight matrix by $\boldsymbol{W}(0)$ and the updated weight matrix at time $t$ by $\boldsymbol{W}(t)$, respectively. We further denote the non-ascending singular values of $\boldsymbol{W}$ at time $t$ by $\{\sigma_i^{\boldsymbol{W}}(t)\}_{i=1}^D$. Then the relative value of the smaller eigenvalues to the larger ones will decrease as $t$ increases. Formally, $\frac{\sigma_i^{\boldsymbol{W}}(t)}{\sigma_j^{\boldsymbol{W}}(t)} \leq \frac{\sigma_i^{\boldsymbol{W}}(t')}{\sigma_j^{\boldsymbol{W}}(t')}, \ \forall t < t', i \leq j$. Furthermore, if the following condition holds: $\lambda_1^P \geq \cdots \geq \lambda_d^P > \lambda_{d+1}^P \geq \cdots \geq \lambda_D^P$, then*

$$\lim_{t \to \infty} \frac{\sigma_i^{\boldsymbol{W}}(t)}{\sigma_j^{\boldsymbol{W}}(t)} = 0, \ \forall i \leq d \text{ and } j \geq d+1. \quad (3)$$

See proof in Appendix A.1. Lemma 1 indicates that the singular values of $\boldsymbol{W}$ (in proportional to the larger ones) shrink as the training step increases. With Lemma 1, we can conclude the following theorem that reveals a dimensional collapse phenomenon under this condition:

**Theorem 1.** (Laplacian regularization leads to dimensional collapse.) *For the linear model above optimized with Graph Laplacian Regularization, the embedding space of nodes tends to be dominated by the largest a few eigenvalues. Specifically, if the covariance matrix of input features is an identity matrix, we have:*

$$\lim_{t \to \infty} \frac{\lambda_i^C(t)}{\lambda_j^C(t)} = 0, \ \forall i \leq d \text{ and } j \geq d+1. \quad (4)$$

See proof in Appendix A.2. Theorem 1 reveals that with the effect of Graph Laplacian Regularization, the node embeddings suffer from dimensional collapse.N As the eigenspectrum is dominated by its largest few eigenvalues, the embedding space is narrow, leading to poor robustness and generalization ability when classified with linear classifiers (see more discussions in Appendix E).

Despite the analysis above, the exact evolving dynamics of node embeddings should be more complicated, as 1) the supervised cross-entropy loss forces nodes of different classes to have distinguishable embeddings; 2) the encoder is usually an MLP with non-linear activations instead of a simple linear model. Therefore we further provide empirical results to show that dimensional collapse does exist.

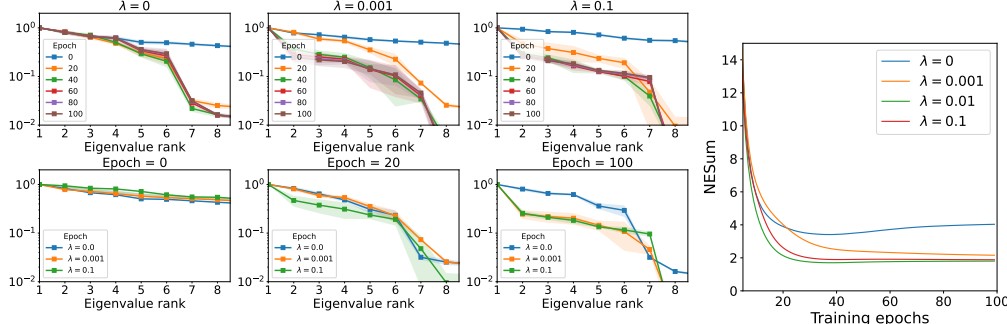

Figure 2: Eigenspectra for node embeddings with different strengths of Laplacian regularization $\lambda$ (the upper three figures), at different training epochs (the lower three figures). x-axis represents the index of sorted eigenvalues and y-axis is the normalized eigenvalue (the ratio to the largest one). The results are averaged over 10 random initialization with 95% confidence intervals.

Figure 3: Evolving of NESum as the training epoch increases, with different regularization factors.

**Empirical justification.** We train a 3-layer MLP model using Eq. 1 on `Cora` dataset. The embedding dimension is set as $512$ and the regularization works on the output of the penultimate layer. To study the dimensional collapse phenomenon, we plot the decay of the eigenvalues of node embeddings' correlation matrix with different strengths of regularization as the training epochs increase in Fig. 2 (here we only plot the top-8 eigenvalues for better visualization, while deferring the original results in Appendix C.4), from which we can easily see that: without Laplacian regularization, the top-$C$ largest eigenvalues (where $C$ is the number of classes, and $C = 7$ for `Cora`) are easily preserved by the model. This means the model learns to discriminate between different classes. However, when we increase the Laplacian regularization factor $\lambda$, we notice an increasing decay rate of top eigenvalues. As the top eigenvalues indicate the realistic embedding dimension that takes effect, we conclude that a large decay rate degrades the importance of the spaces induced by other eigenvalues, thus leading to feature collapse issues.

Besides, we employ *normalized eigenvalue sum* (NESum) introduced in He & Ozay (2022) as a metric to measure the extent of dimensional collapse. Formally, NESum is defined as the ratio between the summation of eigenvalue and the largest one: $\text{NESum}(\{\lambda_i^C\}) \triangleq \sum_{i=1}^d \lambda_i^C / \lambda_1^C$. Intuitively, a large NESum value indicates that the eigenvalues are fluently distributed, while a very small one indicates the dimensional collapse phenomenon (the largest eigenvalue becomes dominant).

In Fig. 3, we plot the evolution of NESum with different regularization strengths. It is observed that 1) NESum decreases as training goes on because the model learns to pay more attention to important features for downstream classification tasks. 2) NESum trained with purely cross-entropy loss converges to a high value., which is because the top-$C$ eigenvalues are preserved. 3) With additional Laplacian regularization, NESum decreases quickly and converges to a small value even if the regularization factor $\lambda$ is small. The above observations demonstrate that Laplacian regularization leads to a larger decay rate of top eigenvalues. The significant decay rate will make the learned representations less informative as the model focuses much more on the dominant eigenvalue rather than equal to the top eigenvalues.

## 4 PROPOSED REMEDY: OVERCOMING DIMENSIONAL COLLAPSE VIA ORTHOGONALITY REGULARIZATION

### 4.1 EXPLICIT REGULARIZATION ON THE CORRELATION MATRIX

Our thorough analysis in Sec. 3 reveals that the poor performance of GR-MLPs could be attributed to less-expressive node representations (due to dimensional collapse). Specifically, we establish that the eigenspectrum of the embeddings' correlation matrix is dominated by the largest eigenvalue (different dimensions are **over-correlated**).

In contrast to dimensional collapse, whitened representations have an identity correlation matrix with equally distributed eigenvalues. Motivated by this, a natural idea should be enforcing a soft regularization term on the correlation matrix of node embeddings, e.g., minimizing the distance between $C$ and the identity matrix $I$:

$$\ell_{corr\_reg} = \|C - I\|_F^2 = \sum_{i=1}^{d}(1 - C_{ii})^2 + \sum_{i \neq j} C_{ij}^2 = \sum_{i \neq j} C_{ij}^2. \qquad (5)$$

Note that the on-diagonal terms $C_{ii} = 1$ for all $i$, so Eq. 5 is essentially forcing the off-diagonal terms of the correlation matrix to become zero, or in other words, making the embeddings **orthogonal**, so that different dimensions of node embeddings can capture orthogonal information. One may directly equip Eq. 5 with existing GR-MLPs for alleviating the dimensional collapse issue. However, we would like to design a more general, flexible, and elegant formulation that can handle high-order connectivity and non-homophily graphs (Pei et al., 2020; Zheng et al., 2022). We then introduce ORTHO-REG, a powerful and flexible GR-MLP model, step by step.

## 4.2 GRAPH-REGULARIZED MLP WITH ORTHO-REG

Similar to previous GR-MLPs, we first use an MLP encoder to map raw node features to the embeddings. This process can be formulated as $H = \mathrm{MLP}_\theta(X)$, where $X = \{x_i\}_{i=1}^{|\mathcal{V}|}$ is raw node features while $H = \{h_i\}_{i=1}^{|\mathcal{V}|}$ is the embedding matrix.

The next question is what kind of graph structure information is more beneficial. Previous GR-MLPs either resort to edge-centric smoothing (Zhou et al., 2003; Ando & Zhang, 2006) or node-centric matching (Yang et al., 2021; Hu et al., 2021). While recent studies indicate that the node-centric method is more appropriate for node-level tasks as edge-centric methods overemphasize the ability to recover the graph structure (Yang et al., 2021). Inspired by this, we employ a **neighborhood abstraction** operation to summarize the neighborhood information as guidance of the central node. Formally, for a node $i \in \mathcal{V}$ and the embeddings of its (up-to) $T$-hop neighbors $\{h_j\}^{(1:T)}(i)$, we can get the summary if its $T$-hop neighborhoods through a pooling function $s_i = \mathrm{Pool}(\{h_j\}^{(1:T)}(i))$. The exact formulation of the pooling function could be flexible to fit graphs with different properties. However, here we consider a simple average pooling of node embeddings from different order's neighborhoods for simplicity, which can work in most cases:

$$S = \sum_{t=1}^{T} \tilde{A}^t H / L, \text{ where } \tilde{A} = A D^{-1}. \qquad (6)$$

To make the node embeddings aware of structural information, we employ the following regularization term on the cross-correlation matrix of node embeddings $H$ and summary embeddings $S$:

$$\mathcal{L}_{reg} = -\alpha \sum_{k=1}^{D} C_{kk} + \beta \sum_{k \neq k'} C_{kk'}^2, \qquad (7)$$

where $C = \{C_{kk'}\} \in \mathbb{R}^{D \times D}$ is the cross-correlation matrix of $H$ and $S$. We show in the following theorem that with Eq. 7, the node embeddings will be locally smoothed and at the same time, prevent dimensional collapse:

**Theorem 2.** *Assume $T = 1$ and $H$ are free vectors. Let $H^*$ be a global optimizer of Eq. 7, then $H^*$ is smoothed over the graph structure and is orthogonal.*

See proof in Appendix A.3. Finally, we can employ an additional linear layer to make predictions $Y = \mathrm{LIN}_\phi(H)$. Then the final objective function to be optimized is:

$$\mathcal{L} = \ell_{xent}(Y^L, \hat{Y}^L) - \alpha \sum_{k=1}^{D} C_{kk} + \beta \sum_{k \neq k'} C_{kk'}^2, \qquad (8)$$

where $\alpha, \beta$ are trade-off hyperparameters to balance the strengths of regularization.

**Remark 2.** *With a well-trained model, we can give prediction for an upcoming node with feature $x$ with $y = \mathrm{Lin}_\phi(\mathrm{MLP}_\theta(x))$ quickly, and without the help of graph structure.*

Table 1: Prediction accuracy of semi-supervised node classification tasks on the seven benchmark graphs. ORTHO-REG outperforms powerful GNN models and competitive MLP-architectured baselines on 6 out of 7 datasets.

| | Methods | Cora | Citeseer | Pubmed | Computer | Photo | CS | Physics |
|---|---|---|---|---|---|---|---|---|
| GNNs | SGC | 81.0±0.5 | 71.9±0.5 | 78.9±0.4 | 80.6±1.9 | 90.3±0.8 | 87.9±0.7 | 90.3±1.4 |
| | GCN | 82.2±0.5 | 71.6±0.4 | 79.3±0.3 | 82.9±2.1 | 91.8±0.6 | 89.9±0.7 | 91.9±1.2 |
| | GAT | 83.0±0.7 | 72.5±0.7 | 79.0±0.3 | 82.5±1.6 | 91.4±0.8 | 90.5±0.8 | 92.3±1.5 |
| KD-MLPs | GLNN | 82.6±0.5 | 72.8±0.4 | 80.2±0.6 | 82.1±1.9 | 91.3±1.0 | 92.6±1.0 | **93.3±0.5** |
| GR-MLPs | MLP | 59.7±1.0 | 57.1±0.5 | 68.4±0.5 | 62.6±1.8 | 76.2±1.4 | 86.9±1.0 | 89.4±0.7 |
| | Lap-Reg | 60.3±2.5 | 58.6±2.4 | 68.7±1.4 | 62.6±2.0 | 76.4±1.1 | 87.9±0.6 | 89.5±0.5 |
| | P-Reg | 64.4±4.5 | 61.1±2.1 | 72.3±1.7 | 68.9±3.3 | 79.7±3.7 | 90.9±1.9 | 91.6±0.7 |
| | GraphMLP | 79.5±0.6 | 73.1±0.4 | 79.7±0.4 | 79.3±1.7 | 90.1±0.5 | 90.3±0.6 | 91.6±0.8 |
| | N2N | 83.2±0.4 | 73.3±0.5 | 80.9±0.4 | 81.4±1.6 | 90.9±0.7 | 91.5±0.7 | 91.8±0.7 |
| Ours | ORTHO-REG | **84.7±0.4** | **73.5±0.4** | **82.8±0.5** | **83.7±1.5** | **92.3±1.0** | **92.9±1.1** | 92.8±0.8 |

## 5 EXPERIMENTS

In this section, we conduct experiments to evaluate ORTHO-REG by answering the following research questions:

- **RQ1**: What's the performance of ORTHO-REG on common transductive node classification tasks compared with GNN models and other MLP models? (Sec. 5.2)
- **RQ2**: On cold-start settings where we do not know the connections of testing nodes, can ORTHO-REG demonstrate better performance than other methods? (Sec. 5.3)
- **RQ3**: Does ORTHO-REG mitigate the dimensional collapse issue, and is each design of ORTHO-REG really necessary to its success? (Sec. 5.4)
- **RQ4**: Can ORTHO-REG demonstrates better robustness against structural perturbations compared with Graph Neural Networks? (Sec. 5.5)

Due to space limits, we defer the experiments on heterophily graphs and scalability comparison in Appendix C.2 and Appendix C.3, respectively. A brief introduction of the baselines is given in Appendix B.3.

### 5.1 EXPERIMENT SETUPS

**Datasets.** We consider 7 benchmark graph datasets and their variants in this section: Cora, Citeseer, Pubmed, Amazon-Computer, Amazon-Photo, Coauthor-CS, and Coauthor-Physics as they are representative datasets used for semi-supervised node classification (Kipf & Welling, 2017; Hu et al., 2021; Zhang et al., 2021; Zheng et al., 2021). The detailed introduction and statistics of them are presented in Appendix B. To evaluate ORTHO-REG on large-scale graphs, we further consider two OGB datasets (Hu et al., 2020): Ogbn-Arxiv and Ogbn-Products. Note that the two OGB datasets are designed for fully-supervised node classification tasks, so we defer their results to Appendix C.

**Implementations.** If not specified, we use a two-layer MLP model as the encoder to generate node embeddings, then another linear layer takes node embeddings as input and outputs predicted node labels. We use Pytorch to implement the model and DGL (Wang et al., 2019) to implement the neighborhood summarizing operation in Eq. 6. If not specified, all our experiments are conducted on an NVIDIA V100 GPU with 16G memory with Adam optimizer (Kingma & Ba, 2015).

### 5.2 TRANSDUCTIVE SEMI-SUPERVISED NODE CLASSIFICATION (RQ1)

We first evaluate our method on transductive semi-supervised node classification tasks. For comparison, we consider three types of baseline models: 1) Graph Neural Networks (GNNs), including SGC (Wu et al., 2019a), GCN (Kipf & Welling, 2017) and GAT (Velickovic et al., 2018). 2) Representative knowledge distillation (KD-MLP) method GLNN (Zhang et al., 2021). 3) Basic MLP and GR-MLP models, including Laplacian Regularization (Lap-Reg, Zhou et al. (2003), Ando & Zhang

Table 2: Test accuracy on the isolated nodes.

| | Methods | Cora | Citeseer | Pubmed |
|---|---|---|---|---|
| GNNs | GCN | 53.02±1.78 | 47.09±1.38 | 71.50±2.21 |
| | GraphSAGE | 55.38±1.92 | 41.46±1.57 | 69.87±2.13 |
| KD-MLPs | ColdBrew | 58.75±2.11 | 53.17±1.41 | 72.31±1.99 |
| | GLNN | 59.34±1.97 | 53.64±1.51 | 73.19±2.31 |
| GR-MLPs | MLP | 52.35±1.83 | 53.26±1.41 | 65.84±2.08 |
| | GraphMLP | 59.32±1.81 | 53.17±1.48 | 72.33±2.11 |
| | ORTHO-REG (Ours) | **61.93±1.77** | **56.31±1.54** | **73.42±1.99** |

Table 3: Effects of different components of ORTHO-REG

| Variants | Cora | Citeseer | Pubmed |
|---|---|---|---|
| Baseline | 84.7 | 73.5 | 82.8 |
| $\alpha = 0$ | 54.7 | 51.4 | 47.2 |
| $\beta = 0$ | 79.3 | 68.7 | 76.8 |
| $T = 1$ | 83.9 | 72.9 | 82.1 |
| $T = 2$ | **84.7** | **73.5** | **82.8** |
| $T = 3$ | 84.3 | 73.3 | 82.5 |

(2006)), Propagation Regularization (P-Reg, Yang et al. (2021)), GraphMLP (Hu et al., 2021), and Node-to-Neighborhood Mutual Information Maximization (N2N, Dong et al. (2022))

For each dataset, we use 20 nodes per class for training, 500 nodes for validation, and another 1000 nodes for testing. For Cora, Citeseer, and Pubmed we use the public split, while for the remaining datasets, we split randomly. We report the average prediction accuracy with standard deviation over 20 random trials in Table 1.

As demonstrated in the table, ORTHO-REG outperforms previous GR-MLPs by a large margin, which greatly validates the importance and effectiveness of orthogonal node embeddings. Compared with the competitive knowledge distillation method GLNN, ORTHO-REG also demonstrates better performance on 6 out of 7 graphs. It is also worth noting that our method even outperforms powerful GNN models such as GCN and GAT, which indicates that node features of the graphs are less exploited by these GNN models. In contrast, our method can fully exploit the potential of node features.

## 5.3    INDUCTIVE NODE CLASSIFICATION FOR COLD-START SCENARIOS (RQ2)

To evaluate the performance of ORTHO-REG on cold-start scenarios where the connections between newly encountered nodes and existing nodes are missing, we follow the setups in ColdBrew that selects a proportion of nodes as *isolated* nodes which will be removed from the original graph. Then the model is evaluated on the isolated nodes in the testing set. Due to the space limit, we present the detailed setups and evaluation methods in Appendix B.2. Besides the baselines used in Zheng et al. (2021), we also include GLNN for a fair comparison.

In Table. 2, we report the experimental results of ORTHO-REG and baseline methods on the isolation nodes. As demonstrated in the table, for isolated nodes whose connectivity in the graph is unknown, GNN models perform poorly as they require both the node features and graph structure for accurate inference. By contrast, MLP-based models generalize better on isolated nodes as they make the best of the available node features. The proposed ORTHO-REG outperforms both GNNs and MLPs (including KD MLPs and GR-MLPs) baselines.

## 5.4    STUDIES OF ORTHO-REG (RQ3)

### 5.4.1    DOES ORTHO-REG MITIGATE DIMENSIONAL COLLAPSE?

In Sec. 3 we have attributed the limitation of previous GR-MLPs to the dimensional collapse phenomenon, and in Sec. 4.2 we have proposed ORTHO-REG to mitigate such a problem from a theoretical perspective. In this part, we would like to empirically show that ORTHO-REG can avoid the dimensional collapse issue by keeping node embeddings' eigenspectra.

In consistency with the settings in Sec. 3, we evaluate the embeddings learned from ORTHO-REG at different training epochs (we take both Cora and Pubmed for illustrations). The decay of eigenvalues of node embeddings' correlation matrix at different epochs is plotted in Fig. 4 (a) and (c). It is observed that the top eigenvalues are well-reserved thanks to the explicit regularization of node embeddings' correlation matrix. In Fig. 4 (b) and (d) we also plot the change of testing accuracy as well as the NESum value as the training epoch increases, from which we could observe a positive relationship between the NESum value and the test accuracy: neglecting the initial oscillations, we notice the test accuracy will grow smoothly as the NESum value increases and will reach its peak

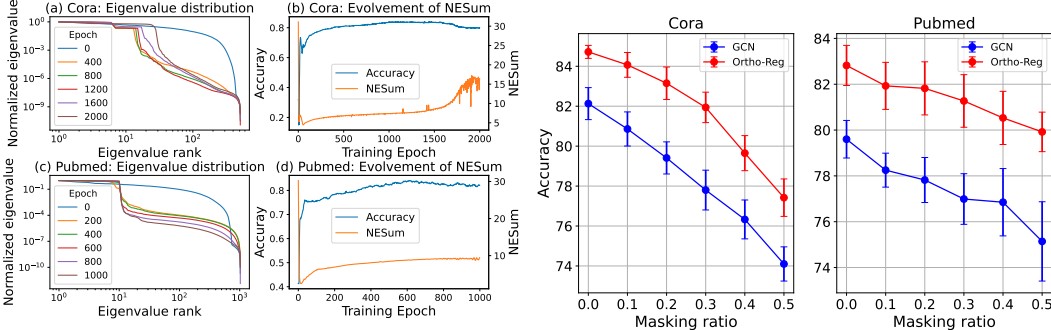

Figure 4: Visualization of ORTHO-REG's impact on node embeddings' Eigenspectra on `Cora` and `Pubmed`.

Figure 5: Performance with different strengths of edge masking ratios.

when NESum overwhelms (`Cora`) or converges (`Pubmed`). The above observations demonstrate that ORTHO-REG does mitigate the dimensional collapse problem and lead to a more powerful model.

### 5.4.2 ABLATION STUDIES

We then conduct ablation studies to study the effect of different components of ORTHO-REG, and we present the results in Table 3. We first study the impact of the two regularization terms by setting the corresponding factors ($\alpha$ and $\beta$) to 0, respectively. When $\alpha = 0$ (i.e., only decorrelating different dimensions), we observe that the model's performance is even worse than the pure MLP model (see in Table 1). This indicates that adding orthogonal regularization is not always beneficial (e.g., for vanilla MLP), but is indeed beneficial for GR-MLPs. By contrast, without orthogonal regularization (i.e., $\beta = 0$), the power of structure regularization is restricted, and decorrelating different dimensions can boost performance greatly. We further investigate whether considering a larger neighborhood would improve the model's performance. The empirical results demonstrate that considering a larger neighborhood improves the performance compared to only using first-order neighborhoods, but $T = 2$ is already optimal for most datasets.

### 5.5 ROBUSTNESS AGAINST STRUCTURAL PERTURBATIONS (RQ4)

Finally, we study the robustness of ORTHO-REG against attacks on the graph structures compared with GNN models. As ORTHO-REG uses node features rather than a combination of node features and edges for prediction, we expect it to demonstrate better robustness under mild structural perturbations. To reach this target, we randomly mask a fraction of the edges of the graph and evaluate the performance of ORTHO-REG and GCN under different edge-masking ratios. In Fig. 5, we plot how the model's performance changes (with standard deviation) as the masking ratio increases with 20 random trials. As demonstrated in Fig. 5, our method demonstrates better robustness against moderate-level edge perturbations. This is because we do not explicitly use the graph structure for generating predictions, making ORTHO-REG less sensitive to perturbations on the graph structure.

## 6 CONCLUSIONS

In this paper, we have proposed ORTHO-REG, a novel Graph-Regularized MLP method for node representation learning. We show that simple graph regularization methods can cause dimensionally collapsed node embeddings both theoretically and empirically. We show that the proposed ORTHO-REG, which enforces the orthogonality of the correlation matrix of node embeddings, can naturally avoid the feature collapse phenomenon. We have conducted extensive experiments, including traditional transductive semi-supervised node classification tasks and inductive node classification for cold-start nodes, demonstrating the superiority of ORTHO-REG.

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

# A    PROOFS

## A.1    PROOFS FOR LEMMA 1

*Proof.* First, let's take the gradient of the regularization loss $\mathcal{L}_{reg}$ with respect to the weight matrix $\boldsymbol{W}$:

$$
\begin{aligned}
\frac{\partial \mathcal{L}_{reg}}{\partial \boldsymbol{W}} &= \frac{\partial \operatorname{tr}(\boldsymbol{H}^\top \boldsymbol{L} \boldsymbol{H})}{\partial \boldsymbol{W}} \\
&= \frac{\partial \operatorname{tr}((\boldsymbol{X}\boldsymbol{W})^\top \boldsymbol{L}(\boldsymbol{X}\boldsymbol{W}))}{\partial \boldsymbol{W}} \\
&= 2\boldsymbol{X}^\top \boldsymbol{L} \boldsymbol{X} \boldsymbol{W} \\
&= 2\boldsymbol{P} \boldsymbol{W}
\end{aligned} \tag{9}
$$

Treat the weight matrix as a function of the training step $t$, i.e., $\boldsymbol{W} = \boldsymbol{W}(t)$, then we can derive the gradient of $\boldsymbol{W}(t)$ with respect to $t$ by $\frac{\mathrm{d}\boldsymbol{W}(t)}{\mathrm{d}t} = 2\boldsymbol{P}\boldsymbol{W}$. As both $\boldsymbol{X}$ and $\boldsymbol{L}$ are fixed, we can solve the equation analytically,

$$
\boldsymbol{W}(t) = \exp(\boldsymbol{P}t) \cdot \boldsymbol{W}(0). \tag{10}
$$

As we have the non-ascending eigenvalues of $\boldsymbol{P}$ as $\lambda_1^P \geq \lambda_2^P \geq \cdots \geq \lambda_D^P$, we can define an auxiliary function $f(t; \lambda_i^P, \lambda_j^P) = \exp(\lambda_i^P t)/\exp(\lambda_j^P t) = e^{(\lambda_i^P - \lambda_j^P)t}$. It is obvious that $f(t; \lambda_i^P, \lambda_j^P)$ is monotonically decreasing for all $j \geq i$. As $\boldsymbol{W}(t)$ is a transformation of its initial state $\boldsymbol{W}(0)$ up to $\exp(\boldsymbol{P}t)$, we can easily conclude that

$$
\frac{\sigma_i^{\boldsymbol{W}}(t)}{\sigma_j^{\boldsymbol{W}}(t)} \leq \frac{\sigma_i^{\boldsymbol{W}}(t')}{\sigma_j^{\boldsymbol{W}}(t')}, \quad \forall \, t < t' \text{ and } i \leq j. \tag{11}
$$

If we further have the condition that $\lambda_1^P \geq \cdots \geq \lambda_d^P > \lambda_{d+1}^P \geq \cdots \geq \lambda_D^P$, we have $\lim_{t \to \infty} f(t; \lambda_i^P, \lambda_j^P) = 0, \forall i \leq d, j \geq d+1$. Then we are able to complete the proof. $\qquad \square$

## A.2    PROOFS FOR THEOREM 1

*Proof.* The embedding space is identified by the eigenspectrum of the correlation (covariance) matrix $\boldsymbol{C}$ of node embeddings $\boldsymbol{H}$. As $\boldsymbol{H} = \boldsymbol{X}\boldsymbol{W}$, its correlation matrix can be (simply) identified as:

$$
\begin{aligned}
\boldsymbol{C} &= \sum_{i=1}^{N} (\boldsymbol{h}_i - \overline{\boldsymbol{h}})^\top (\boldsymbol{h}_i - \overline{\boldsymbol{h}})/N \\
&= \sum_{i=1}^{N} \boldsymbol{W}^\top (\boldsymbol{x}_i - \overline{\boldsymbol{x}})^\top (\boldsymbol{x}_i - \overline{\boldsymbol{x}})\boldsymbol{W}/N.
\end{aligned} \tag{12}
$$

According to Lemma 1, $\boldsymbol{W}$ has shrinking singular values, so $\boldsymbol{C}$ has vanishing eigenvalues, indicating collapsed dimensions.

Specially, when the input features have an identity matrix, we have:

$$
\begin{aligned}
\boldsymbol{C} &= \sum_{i=1}^{N} \boldsymbol{W}^\top (\boldsymbol{x}_i - \overline{\boldsymbol{x}})^\top (\boldsymbol{x}_i - \overline{\boldsymbol{x}})\boldsymbol{W}/N. \\
&= \boldsymbol{W}^\top \sum_{i=1}^{N} \frac{(\boldsymbol{x}_i - \overline{\boldsymbol{x}})^\top (\boldsymbol{x}_i - \overline{\boldsymbol{x}})}{N} \boldsymbol{W} \\
&= \boldsymbol{W}^\top \boldsymbol{W}.
\end{aligned} \tag{13}
$$

$\qquad \square$

Thus, for $\boldsymbol{C}$'s eigenvalues $\{\lambda_i^C\}_{i=1}^D$, we have $\lambda_i^C = (\sigma_i^{\boldsymbol{W}})^2$. Then Theorem 1 can be easily concluded with Lemma 1.

### A.3 PROOFS FOR THEOREM 2

*Proof.* Note that we aim to optimize the following objective function (Eq. 7):

$$\mathcal{L} = -\alpha \sum_{k=1}^{D} C_{kk} + \beta \sum_{k \neq k'} C_{kk'}^2.$$

As the first term applies only on the on-diagonal terms of the correlation matrix and the second term applies only on the off-diagonal terms, we are able to study the effects of two terms respectively:

$$
\begin{aligned}
\mathcal{L}_{on-diag} &= -\alpha \sum_{k=1}^{D} C_{kk} \\
\mathcal{L}_{off-diag} &= \beta \sum_{k \neq k'}^{D} C_{kk'}^2
\end{aligned}
\tag{14}
$$

For the on-diagonal terms $\mathcal{L}_{on-diag}$, as we have set $T = 1$, we have $\boldsymbol{C} = \sum_{i=1}^{N} \boldsymbol{h}_i \boldsymbol{s}_i^\top / N$, and $C_{kk} = \sum_{i=1}^{N} (h_i)_k \cdot (s_i)_k / N$ (the subscript $k$ denotes the $k$-th dimension). Then,

$$\frac{\partial C_{kk}}{\partial (h_i)_k} = \frac{1}{N}(s_i)_k, \text{ and } \frac{\partial C_{kk}}{\partial (h_i)'_k} = 0, \ \forall k' \neq k. \tag{15}$$

As a result,

$$\frac{\partial \sum_{k=1}^{D} C_{kk}}{\partial \boldsymbol{h}_i} = \frac{1}{N}\boldsymbol{s}_i = \frac{1}{N}\frac{\sum_{j \in \mathcal{N}(i)} \boldsymbol{h}_j}{|\mathcal{N}(i)|}. \tag{16}$$

Eq. 16 indicates that the on-diagonal terms force each node embedding to be smoothed within its first-order neighborhoods.

Then we turn to the off-diagonal terms $\mathcal{L}_{off-diag}$. Similarity, we have $C_{kk'} = \sum_{i=1}^{N} (h_i)_k \cdot (s_i)_{k'} / N$. As for both $\boldsymbol{H}$ and $\boldsymbol{S}$, the diagonal term of their correlation matrixes for each dimension should be equal to $1$, formally,

$$\frac{\sum_{i=1}^{N} (h_i)_k^2}{N} = \frac{\sum_{i=1}^{N} (s_i)_k^2}{N} = 1. \tag{17}$$

Then according to Cauchy–Schwarz inequality, we have:

$$
\begin{aligned}
\left[\sum_{i=1}^{N}(h_i)_k \cdot (s_i)_k\right]^2 &\leq \left[\sum_{i=1}^{N}(h_i)_k^2\right]\left[\sum_{i=1}^{N}(s_i)_k^2\right] = N^2 \\
\frac{\sum_{i=1}^{N}(h_i)_k \cdot (s_i)_k}{N} &\leq 1
\end{aligned}
\tag{18}
$$

and the equality holds if and only if $(h_i)_k = (s_i)_k, \forall i$. As a result, the global optimizer of $\mathcal{L}_{on-diag}$ will induce $\boldsymbol{h}_i = \boldsymbol{s}_i, \forall i$. Then, when $C_{kk'} = 0, k \neq k'$, we have:

$$\sum_{i=1}^{N}(h_i)_k \cdot (s_i)_{k'}/N = \sum_{i=1}^{N}(h_i)_k \cdot (h_i)_{k'}/N = C_{kk'}^{auto} = 0, \tag{19}$$

where $\boldsymbol{C}^{auto}$ is the auto-correlation matrix of $\boldsymbol{H}$. So we've completed the proof. $\qquad \square$

# B    Experiment Details in Section 5

## B.1    Datasets Details

We present the statistics of datasets used in transductive node classification tasks in Table 4, including the number of nodes, number of edges, number of classes, number of input node features as well as the number of training/validation/testing nodes.

Table 4: Statistics of benchmarking datasets in transductive settings

| Dataset | #Nodes | #Edges | #Classes | #Features | #Train/Val/Test |
|---|---|---|---|---|---|
| Cora | 2,708 | 10,556 | 7 | 1,433 | 140/500/1000 |
| Citeseer | 3,327 | 9,228 | 6 | 3,703 | 120/500/1000 |
| Pubmed | 19,717 | 88,651 | 3 | 500 | 60/500/1000 |
| Coauthor-CS | 18,333 | 327,576 | 15 | 6,805 | 300/500/1000 |
| Coauthor-Physics | 34,493 | 991,848 | 5 | 8,451 | 100/500/1000 |
| Amazon-Computer | 13,752 | 574,418 | 10 | 767 | 200/500/1000 |
| Amazon-Photo | 7,650 | 287,326 | 8 | 745 | 160/500/1000 |

For the inductive node classification tasks in cold-start settings, we also present the statistics of Cora, Citeseer and Pubmed in Table 5, where we provide the number of isolated nodes, the number of tail nodes as well as the number of edges left after removing the isolated nodes from the graph.

Table 5: Statistics of benchmarking datasets in inductive settings

| Dataset | #Nodes | #Edges | #Isolated | #Tail | #Edges left |
|---|---|---|---|---|---|
| Cora | 2,708 | 10,556 | 534 | 534 | 9,516 |
| Citeseer | 3,327 | 9,228 | 676 | 676 | 7,968 |
| Pubmed | 19,717 | 88,651 | 4,547 | 4,547 | 79,557 |

## B.2    Details of Experiments on Cold-Start Scenarios (Sec. 5.3)

In this section, we detailedly elaborate how the inductive isolated nodes are selected. Note that our processing directly follows the officially-implemented codes[1] in ColdBrew (Zheng et al., 2021). For each dataset, we first rank among the nodes according to their node degrees, through which we are able to get the degree of the bottom 3th percentile node, termed by $d_{3th}$. Then we screen out nodes whose degree is smaller than or equal to $d_{3th}$ as isolated nodes, which are subsequently removed from the original graph.

Note that in these datasets most of nodes only have a few connections (e.g. 1 or 2), the actually numbers of isolated nodes and tail nodes are usually much larger than the expected 3%. See Table 5 for details.

For each dataset, we use the fixed 20 nodes per class (as in the public split) for training and all the remaining nodes for testing.

## B.3    Introduction of baselines

In Sec. 5.1 we've briefly introduced the baselines for comparison, here we'd like to detailedly introduce the MLP-based baselines.

**KD-MLPs:**  We have covered two KD-MLP models: GLNN (Zhang et al., 2021) and Cold-Brew (Zheng et al., 2021).

---

[1]https://github.com/amazon-research/gnn-tail-generalization

- **GLNN**: As a typical GNN-to-MLP knowledge distillation method, given the predicted soft labels from a well-learned GNN model $\{z_i\}$, GLNN learns an MLP model through jointly optimizing the supervised loss on labeled nodes and the cross-entropy loss between MLP's predictions and GNN's predictions over all nodes:

$$\mathcal{L}_{glnn} = \mathcal{L}_{sup} + \lambda \mathcal{L}_{kd}$$
$$\mathcal{L}_{sup} = \sum_{i \in \mathcal{V}^L} \boldsymbol{\ell}_{xent}(\boldsymbol{y}, \hat{\boldsymbol{y}}) \text{ and } \mathcal{L}_{kd} = \sum_{i \in \mathcal{V}} \mathcal{D}_{KL}(\hat{\boldsymbol{y}}_i, \boldsymbol{z}_i), \quad (20)$$

where $\lambda$ is a trade-off hyperparameter.

- **ColdBrew**: As another KD-MLP model, ColdBrew is specially designed to handle cold-start problems and has a totally different formulation compared with GLNN. First, it equips the teacher GNN model with structural embedding so that it can overcome the oversmoothing issue. Then, besides the knowledge distillation loss, it discovers the virtual neighborhood of each node using the embedding learned from the student MLP. With this operation, the model is able to estimate the possible neighbors of each node and thus can generalize better in inductive cold-start settings.

**GR-MLPs:** We then introduce the covered GR-MLP models, including Lap-Reg (Zhou et al., 2003; Ando & Zhang, 2006), P-Reg (Yang et al., 2021) and GraphMLP (Hu et al., 2021) detailedly. Besides the basic supervised cross-entropy loss, GR-MLPs employ a variety of regularization losses to inject the graph structure knowledge into the learning of MLPs implicitly.

- **Lap-Reg**: Based on the graph homophily assumption, Lap-Reg enforces Laplacian smoothing on the predicted node signals over the graph structure. Its regularization target could be formulated as:

$$\mathcal{L}_{lap-reg} = \text{tr}(\boldsymbol{Y}^\top \boldsymbol{L} \boldsymbol{Y}), \quad (21)$$

where $\boldsymbol{Y}$ the predicted node signals and $\boldsymbol{L}$ is the Laplacian matrix of the graph.

- **P-Reg**: Similar to Lap-Reg, P-Reg is also built on top of the graph homophily assumption. However, instead of using edge-centric smoothing regularization, P-Reg employs a node-centric proximity preserving term that maximizes the similarity of each node and the average of its neighbors. The regularization objective could be formulated as:

$$\mathcal{L}_{P-reg} = \frac{1}{N} \phi(\boldsymbol{H}, \tilde{\boldsymbol{A}}\boldsymbol{H}), \quad (22)$$

where $\tilde{\boldsymbol{A}}\boldsymbol{H}$ is the propagated node embedding matrix, and $\phi$ is a function that measures the difference between $\boldsymbol{H}$ and $\tilde{\boldsymbol{A}}\boldsymbol{H}$, which could be implemented with a variety of measures like Square Error, Cross Entropy, Kullback-Leibler Divergence, etc.

- **Graph-MLP**: Inspired by the success of contrastive learning, Graph-MLP tries to get rid of GNN models by contrasting between connected nodes. Formally:

$$\mathcal{L}_{graph-mlp} = \frac{1}{N} \sum_{i=1}^{N} - \log \frac{\sum_{j \in \mathcal{N}(i)} \exp(\text{sim}(\boldsymbol{h}_i, \boldsymbol{h}_j)/\tau)}{\sum_{k \in \mathcal{V}} \exp(\text{sim}(\boldsymbol{h}_i, \boldsymbol{h}_k)/\tau)}. \quad (23)$$

The final objective function is also a trade of between the supervised cross-entropy loss and the regularization loss.

## C   ADDITIONAL EXPERIMENTS

### C.1   EXPERIMENTS ON OGB-GRAPHS

To study the effectiveness of ORTHO-REG on large-scale graphs, we conduct experiments on two large-scale graphs: `Ogbn-Arxiv` and `Ogbn-Products`, and we present the results in this section.

The statistics of the two datasets in the transductive setting and the inductive cold-start setting are presented in Table 6 and Table 7. Note that the official split of OGB datasets is different from other datasets in Sec. 5 and does not follow the semi-supervised setting.

Table 6: Statistics of OGB datasets in transductive settings

| Dataset | #Nodes | #Edges | #Classes | #Features | #Train/Val/Test |
|---|---|---|---|---|---|
| Ogbn-Arxiv | 169,343 | 2,332,486 | 40 | 128 | 90,941 / 29,799 / 48,603 |
| Ogbn-Products | 2,449,029 | 123,718,024 | 47 | 100 | 196,615 / 39,323 / 2,213,091 |

Table 7: Statistics of OGB datasets in inductive settings

| Dataset | #Nodes | #Edges | #Isolated | # Nodes left | # Edges left |
|---|---|---|---|---|---|
| Ogbn-Arxiv | 169,343 | 2,332,486 | 16,934 | 152,409 | 2,298,618 |
| Ogbn-Products | 2,449,029 | 123,718,024 | 244,902 | 2,204,127 | 123,661,058 |

For fair comparison, we use the same model size (i.e., the number of parameters) for each model. We train each model for 10 times on each dataset and report the average accuracy with standard deviation on transductive setting and inductive cold-start setting in Table 8 and Table 9 respectively.

Table 8: Test accuracy on OGB datasets in transductive settings.

| | Methods | Ogbn-Arxiv | Ogbn-Products |
|---|---|---|---|
| GNNs | GCN | **71.74±0.29** | 75.26±0.21 |
| | SAGE | 71.49±0.27 | **78.61±0.23** |
| KD-MLPs | GLNN | 69.37±0.25 | 75.19±0.34 |
| GR-MLPs | MLP | 56.28±0.37 | 61.06±0.08 |
| | Lap-Reg | 57.83±0.52 | 65.91±0.31 |
| | P-Reg | 58.41±0.45 | 65.32±0.28 |
| | GraphMLP | 61.11±0.36 | 68.54±0.33 |
| Ours | ORTHO-REG | 70.35±0.22 | 74.35±0.19 |

Table 9: Test accuracy on the isolated nodes of OGB datasets.

| | Methods | Ogbn-Arxiv | Ogbn-Products |
|---|---|---|---|
| GNNs | GCN | 44.51±0.85 | 56.62±1.12 |
| | GraphSAGE | 47.32±0.89 | 57.88±1.01 |
| KD-MLPs | ColdBrew | 52.36±0.84 | 61.64±0.98 |
| | GLNN | 53.18±1.05 | 63.09±0.87 |
| GR-MLPs | MLP | 51.03±0.75 | 60.18±0.84 |
| | Lap-Reg | 51.87±0.81 | 60.47±0.77 |
| | P-Reg | 51.79±0.88 | 60.59±0.91 |
| | GraphMLP | 52.21±0.91 | 61.12±0.98 |
| | ORTHO-REG (Ours) | **54.51±0.77** | **63.95±0.74** |

As demonstrated in Table 8, though under-performing GNN models, ORTHO-REG gives quite good performance (which are very close to that of GNNs) on these two datasets. On inductive cold-start prediction tasks, ORTHO-REG also outperforms both GNN models and other MLP models.

## C.2 EXPERIMENTS ON HETEROPHILY GRAPHS

Then we study the generalization ability of ORTHO-REG on heterophily (non-homophily) graphs. We first give the formal definition of graph homophily ratio as follows:

**Definition 1.** *(Graph Homophily Ratio) For a graph $\mathcal{G} = (\mathcal{V}, \mathcal{E})$ with adjacency matrix $\boldsymbol{A}$, its homophily ratio $\phi$ is defined as the probability that two connected nodes share the same label:*

$$\phi = \frac{\sum_{i,j \in \mathcal{V}} A_{ij} \cdot \mathbb{1}[\boldsymbol{y}_i = \boldsymbol{y}_j]}{\sum_{i,j \in \mathcal{V}} A_{ij}} = \frac{\sum_{i,j \in \mathcal{V}} A_{ij} \cdot \mathbb{1}[\boldsymbol{y}_i = \boldsymbol{y}_j]}{|\mathcal{E}|} \tag{24}$$

The evaluated datasets previously are all homophily graphs, where connected nodes tend to share the same labels. To evaluate ORTHO-REG more extensively we consider three more widely used heterophily graphs: Chameleon, Squirrel and Actor. We provide statistics of the three datasets in Table 10.

For heterophily graphs where the graph homophily assumption does not hold (McPherson et al., 2001; Ciotti et al., 2016), it might be improper to enforce node embeddings/predictions to be smoothed over the graph structure. As a result, we modify the neighborhood abstraction function so that it can better adjust to heterophily graphs. Specifically, we adopt the following function to construct summary embedding:

$$\boldsymbol{S} = \tilde{A}^2 \boldsymbol{H}/T. \tag{25}$$

Compared with Eq. 6, we use only the second neighbors for heterophily graphs. We present the results on the three heterophily graphs in Table 11. For comparison with GNN models, we cover additional

Table 10: Statistics of heterophily graphs

| Dataset | #Nodes | #Edges | #Classes | #Features | Heterophily ratio $\phi$ |
|---|---|---|---|---|---|
| Chameleon | 2,277 | 36,101 | 5 | 2,325 | 0.25 |
| Squirrel | 5,201 | 217,073 | 5 | 2,089 | 0.22 |
| Actor | 7,600 | 33,544 | 5 | 931 | 0.24 |

Table 11: Performance on heterophily graphs

| | Methods | Chameleon | Squirrel | Actor |
|---|---|---|---|---|
| GNNs | Geom-GCN | 67.32±1.76 | 46.01±1.27 | 30.59±0.76 |
| | GPRGNN | 66.31±2.05 | 50.56±1.51 | 30.78±0.83 |
| KD-MLPs | GLNN | 60.58±1.72 | 43.72±1.16 | 34.12±0.77 |
| GR-MLPs | MLP | 47.59±0.73 | 31.67±0.61 | 35.93±0.61 |
| | Lap-Reg | 48.72±1.52 | 30.44±0.97 | 33.71±0.59 |
| | ORTHO-REG (ours) | 63.55±0.83 | 48.72±0.93 | **36.64±0.67** |

two GNN models that specifically designed for handle heterophily graphs: Geom-GCN (Pei et al., 2020) and GPRGNN (Chien et al., 2021)

As demonstrated in the Table, though can't match the performance of advanced GNN models on heterophily graphs, our method greatly narrows the gap between MLPs and GNNs. Specifically, on heterophily graphs where GNNs even perform worse than the vanilla MLP, our model is able to achieve even better performance, thanks to the MLP-based encoder and the regularization loss.

## C.3 SCALABILITY TEST

Finally, we show that with an MLP model as encoder, ORTHO-REG is able to perform inference fast without the reliance on graph structure. We plot the inference time of ORTHO-REG and GraphSAGE on Ogbn-Products with different model depths in Fig. 6. The result demonstrates the superiority of the inference benefit of ORTHO-REG over GNN models.

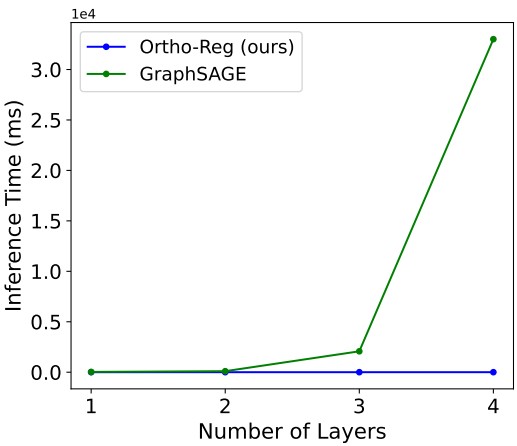

Figure 6: The inference time comparison of GraphSAGE and ORTHO-REG on Ogbn-Products. ORTHO-REG is able to perform inference much faster than GraphSAGE.

## C.4 COMPLETE EMPIRICAL RESULTS FOR SEC. 3

In Fig. 2, we only plot the evolving of top-8 eigenvalues for better visualization. Here, we plot the evolving of all 512 eigenvalues. We also provide the result when $\lambda = 0.01$ for better comparison.

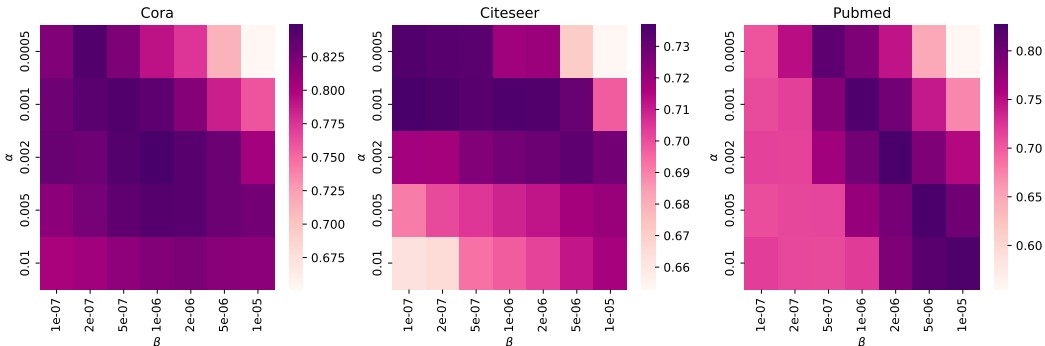

Figure 7: Eigenspectra for node embeddings with different strengths of Laplacian regularization ($\lambda$). The node embeddings are from the second last layer, with a dimension $d = 512$. x-axis represents the index of sorted eigenvalues while y-axis is the corresponding normalized values.

## C.5 SENSITIVITY ANALYSIS OF TRADE-OFF HYPERPARAMETERS

Figure 8: Performance heat map when using different $\alpha$, $\beta$ combinations in Eq. 7, on `Cora`, `Citeseer` and `Pubmed`.

In this section, we study how the two trade-off hyperparameters $\alpha$ and $\beta$ affects the performance of ORTHO-REG. We try different combinations of $\alpha$ and $\beta$ on `Cora`, `Citeseer`, and `Pubmed`, and plot the performance heatmap in Fig. 8.

The conclusion is very interesting: the performance of ORTHO-REG is not very sensitive to a specific value of $\alpha$ or $\beta$. In other words, for a reasonable value of $\alpha$ ($\beta$), we can easily find another value of $\beta$ ($\alpha$) that can achieve similarly high performance. The ratio between $\alpha$ and $\beta$ seems much more important. From Fig. 8, we can observe that for `Cora`, $\alpha/\beta = 2 * 10^3$, and for `Pubmed`, $\alpha/\beta = 1 * 10^3$ can lead to the optimal performance; changing the value of $\alpha$ while fixing $\alpha/\beta$ will not change the performance very much.

## D REPRODUCIBILITY

Please check the supplementary material.

## E RELATIONSHIP BETWEEN DIMENSIONAL COLLAPSE AND LINEAR CLASSIFICATION PERFORMANCE

In Sec. 3, we mainly demonstrate that the dimensional collapse phenomenon does exist in the typical GR-MLP method Laplacian Regularization. In this section, we'd like to use a simple example to explain why dimensional collapse may lead to limited expression power and sub-optimal linear classification performance, which will better support the motivation of this work.

To simplify the analysis, we consider a two-class classification problem where each data point is embedded in 2-dimensional space $\mathcal{R}^2$. The two dimensions are denoted by $e_1$ and $e_2$, respectively. In Fig. 9, we consider three cases: 1) complete dimensional collapse where the embeddings fall on a line

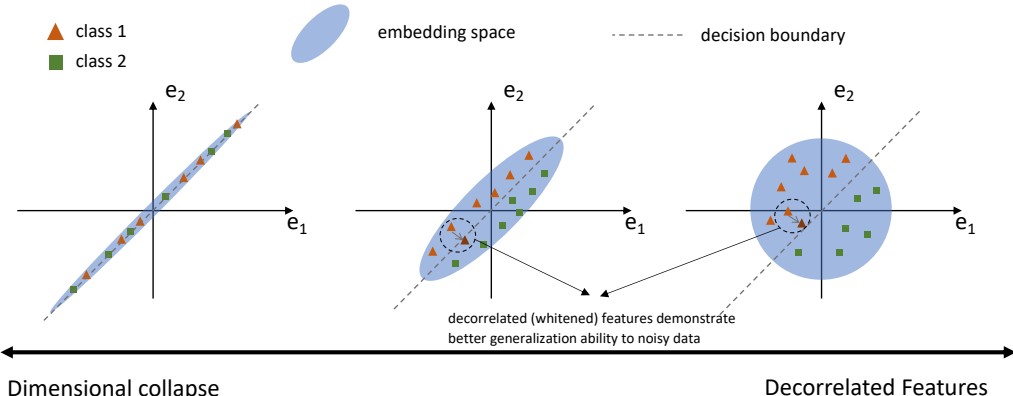

Figure 9: How dimensional collapse affects the performance of linear classification. Left: complete dimensional collapse; Mid: weak dimensional collapse; Right: perfectly decorrelated features.

(the left case); 2) weak dimensional collapse where the variance of the larger eigenvalue's direction is much larger than the other one (the middle case); 3) perfectly decorrelated features where the two eigenvalues' direction are equally important (the right case). We assume that the labels of training data are generated according to $\hat{y} = \text{sgn}(e_2 - e_1)$, i.e., the data point will be labeled as class 1 when $e_2 > e_1$, and class 2 otherwise. As a result, an obvious linear decision boundary will be $e_2 - e_1$.

In the first case (complete dimensional collapse), the data points cannot be linearly classified as they all fall on the same line. In the remaining two cases, data points belonging to two different classes can be easily separated with the decision boundary illustrated above. However, the above results only hold when the data points and labels are generated exactly following the above assumption, which is hard to meet. In many cases, the input data might be noisy, leading to shifting in its final representation. As shown in the circled areas in Fig. 9, when the embedding shift phenomenon occurs (from the brown triangle to the brown triangle) in the weak dimensional collapse case, the shifted embedding can be easily misclassified by the original linear classifier, even if the original embedding is already far from the decision boundary. By contrast, for perfectly decorrelated features, the embeddings can show better tolerance for data noise. Besides, decorrelated features can also show better robustness to attacks and better generalization ability to testing data, thanks to a wider embedding space for each class separated by the decision boundary.

