# OpenReview forum: "OrthoReg: Improving Graph-regularized MLPs via Orthogonality Regularization"
_ICLR.cc/2023/Conference — Submitted to ICLR 2023_

### Official Review · Reviewer_JMnM · 2022-10-19

**Confidence:** 3
**Correctness:** 3
**Technical Novelty And Significance:** 3
**Empirical Novelty And Significance:** 3
**Recommendation:** 6

**Clarity, Quality, Novelty And Reproducibility:**

Clarity
=====
For the most part the submission is clear. I have some questions though:
* Is th $\mathbf{T}$ in the appendix related with the $T$ in equation 5?
* I cannot follow the two last steps in Equation 8. Could you explain them?
* You claim to introduce a new model but the submission is centered on the regularization technique. Are there other model-level contributions besides the regularization?
* Why didn't you choose baselines like the ones in [B]?
* Do you regularize the correlation or the covariance matrix?

There are some minor typos that sometimes hinder readability:
* Section 2.1: However, their performances are still hard to match that of GNN models. -> However, their performances are still hard to match *compared to* that of GNN models.
* Section 2.3: Ineffective -> Inefficient
* Theorem 1: the largest a few -> the largest few
* Section 4.2: while recent -> While recent
* RQ3: mitigates -> mitigate
* Section 5.4.1: lead -> leads
* Section 5.5: rather than -> rather than edges

Quality
=====
The method is sound and the claims are reasonably supported. About the experimental quality:
* There is no much difference between the curves in Figure 2. Are they for one run or multiple runs? Could you provide confidence intervals?
* Sometimes results in bold in the tables have their confidence intervals overlapping with those of other rows. Please either remove the bold font in these cases or put the overlapping numbers in bold too.

Novelty
======
* Besides the overlap with other literature in imposing orthogonality in neural networks, and the name of the method [A], this work is novel to the best of my knowledge.

Reproducibility
===========
* The authors provide the code and ablations.

**Strength And Weaknesses:**

Strengths
=======
* The proposed method is sound and it improves performance of GraphMLPs.
* The authors provide code and ablations, which makes their method more reproducible
* The additional analysis / empirical study is interesting

Weaknesses
=========
* There exists another OrthoReg in the literature [A] that is also a regularization that enforces orthogonality. Although this does not affect the novelty of this work (since they were proposed for different purposes), it could generate some confusion. I suggest to change the name to CorrReg / OrthoGraph / OrthoGraphMLP / OrthoNodeReg / OrthoGraphReg / ....
* The number of baselines is small compared to other works [B]. Is there any reason for that?
* In Figure 2 I cannot see a big difference between different coefficients.

[A]  "Regularizing cnns with locally constrained decorrelations." ICLR 2017.

[B] "Node Representation Learning in Graph via Node-to-Neighbourhood Mutual Information Maximization." Proceedings of the IEEE/CVF Conference on Computer Vision and Pattern Recognition. 2022.

**Summary Of The Paper:**

The authors propose a theoretical and empirical study on the problem of dimensional collapse due to over-smoothing in graph-regularized MLPs. Then, they propose to regularize the cross-correlation between node features and pooled features to alleviate this problem. The resulting model "Ortho-Reg" surpasses the performance of the compared baselines on semi-supervised and cold-start scenarios. Additional experiments show that the proposed regularization helps to alleviate the smoothing problem, that the proposed model is robust to perturbations, and the robustness to different hyperparameter choices.


**Summary Of The Review:**

Overall the method is sound and interesting, the authors add details for reproducibility and include theoretical and empirical insights. Most of my concerns are about clarity and I suggest the authors to go through my questions above. Also, there is the name of the method, which has been used in the past in a slightly-related topic.

Given that this work still needs polishing, I think it is "marginally below the acceptance threshold". However, I think my concerns are easy to address in a rebuttal and I encourage the authors to do so.

After rebuttal
==========
The authors' responses clarified all my questions and resolved most of the other reviewers' concerns. Thus, I raise my score to accept.

---

> ### Author Response · Authors · 2022-11-13
> **Responses to Reviewer JMnM, Batch 1 of 2**
>
> We thank the reviewer for the thorough comments and valuable suggestions. We've fixed the typos and would like to address your concerns by answering your questions one by one:
>
> Q1: Didn't choose the baselines like those in [B].
>
> A1: We thank the reviewer for mentioning [B]. [B] studies a supervised/unsupervised node representation learning task where the encoder is also an MLP. Considering that the supervised model, named N2N(JL) could also be regarded as a GR-MLP model, it is highly related to our work and should also be a baseline. As in [B] the authors use a different experimental setting; we reproduce it according to the authors' codes and have included the results in the revised version (in Table 1). It has to be mentioned that the TAPS sampling strategy is not clearly explained in the paper and is also not presented in the authors' codes (they simply provide the sampled positive examples for Cora dataset), the version we reproduce is N2N (JL) without using TAPS.
>
> For the baselines used in [B], it is good that [B] can use so many baseline methods, but it seems unnecessary as most of these baseline methods are not representative ones. Besides, they perform similarly and much worse than the proposed method. The purpose of using these baselines is to basically show that the proposed N2N model can beat a lot GNN models under the given experimental setting.
>
>  In our paper, we mainly study the performance gap between Graph-Regularized MLPs and GNNs in supervised settings. As a result, we mainly compare our method with other methods using MLP as encoders (e.g., KD-MLPs and GR-MLPs). To show that our method can match the performance of GNNs, we select three representative methods as GNN baselines: SGC, GCN and GAT. We didn't select more advanced GNN models as they might consist of complicated designs (e.g., more parameters or deeper architectures). Note that although we employ a GNN-like module to summarize neighborhood embeddings (i.e., Eq.(5)), it is shallow (i.e., T=2) and parameter-free (it is even simpler than SGC). As a result, it is fair to compare the proposed OrthoReg with the three GNN baselines. Besides, we'd like to mention that our method can even match the performance of SOTA GNN models with complicated regularizations, like GRAND[2]. E.g., on Pubmed, our method gets 82.8\% accuracy, which is higher than 82.7\% of GRAND (see Fig.1).
>
> References:
>
> [1] Wenzheng Feng, Jie Zhang, Yuxiao Dong, Yu Han, Huanbo Luan, Qian Xu, Qiang Yang, Evgeny
> Kharlamov, and Jie Tang. Graph random neural networks for semi-supervised learning on graphs.
> In NeurIPS, 2020.
>
> Q2: Differences between different coefficients in Fig 2.
>
> A2: We are sorry for presenting a figure where the difference between different coefficients is hard to discriminate. The reason that the differences of the eigenspectra as the weight term increases are challenging to determine is that we plot the changes of all $512$ eigenvalues. To better show the change. We replot Fig. 2, where we only focus on the decay of top-8 eigenvalues (we put the original figure to Fig.7 in Appendix C.4 as a reference). Besides, to better show the change of eigenspectum for increasing $\lambda$, we add another three subfigures that plot the top eigenvalues of three different $\lambda$ values at the same training epoch (epoch 0, 20, and 100). As suggested by you, we also plot the 95\% confidence intervals (with ten random initializations). In the updated Fig.2 (especially the lower three subfigures), we can easily observe that as $\lambda$ increases, the ratio of top eigenvalues (w.r.t. the largest one) decreases quicker.

---

> ### Author Response · Authors · 2022-11-13
> **Responses to Reviewer JMnM, Batch 2 of 2**
>
> Q3: The relation of $\mathbf{T}$ in the appendix to the $T$ in Eq.(5).
>
> A3: The two terms are different. $\mathbf{T}$ is a matrix which is defined as $\mathbf{T} = \mathbf{X}^{\top}\mathbf{L}\mathbf{X}$ in Lemma 1, whereas $T$ is defined in Sec. 4.2 denoting $T$-hop neighbors. We are sorry for causing such confusion, and we have replaced $\mathbf{T}$ with $\mathbf{P}$ in the updated version.
>
> Q4: Last two steps of the derivation in Eq.(8).
>
> A4: We already have:
> $$\frac{\partial \mathcal{L}_{reg}}{\partial \mathbf{W}} =  \frac{\partial {\rm tr} ((\mathbf{XW})^{\top}\mathbf{L}(\mathbf{XW}))}{\partial \mathbf{W}} \\\\
> = \frac{\partial {\rm tr} (\mathbf{W}^{\top}\mathbf{X}^{\top}\mathbf{L}\mathbf{XW})}{\partial \mathbf{W}} $$
> Denote $\mathbf{P}=\mathbf{X}^{\top}\mathbf{LX}$, we have:
> $$\frac{\partial {\rm tr} (\mathbf{W}^{\top}\mathbf{X}^{\top}\mathbf{L}\mathbf{XW})}{\partial \mathbf{W}} = \frac{\partial {\rm tr} (\mathbf{W}^{\top}\mathbf{P}\mathbf{W})}{\partial \mathbf{W}} = \mathbf{PW}+\mathbf{P^{\top}W} $$
> As $\mathbf{P}=\mathbf{X}^{\top}\mathbf{LX} = (\mathbf{X}^{\top}\mathbf{LX})^{\top} = \mathbf{P}^{\top}$ is symmetric, we finally have:
> $$\mathbf{PW}+\mathbf{P^{\top}W} = 2\mathbf{PW}$$
> The reviewer's confusion about the previous derivation might be that we use two similar terms $\mathbf{T}$ and $T$ in the initial version. In the latest version, we have replaced $\mathbf{T}$ with $\mathbf{P}$ to avoid confusion.
>
> Q5: Model-level contribution.
>
> A5: We do have model-level contribution besides the regularization term. Our model-level contribution is mainly about the neighborhood abstraction function, i.e., Eq.(6), which introduces an edge-centric operation to capture the (multi-hop) neighborhood information of a target node. This mitigates the issue in existing GR-MLPs that only consider first-order neighborhood information. Besides, our ablation study in Table 3 validates the values of higher-order information.
>
> Q6: Do we regularize the correlations or the covariance matrix?
>
> A6: In our implementation, we regularize the correlation matrix $\mathbf{C}$ instead of the covariance matrix $\mathbf{\Sigma}$. Note that
> $$C_{kk'} = \frac{\Sigma_{kk'}}{\sqrt{\Sigma_{kk}\Sigma_{k'k'}}} (\text{as in Eq.(2)}) $$
>
> So regularizing the covariance matrix should have a similar effect.
>
> Q7: Model name.
>
> A7: We thank the reviewer for mentioning a previous work that uses the same model name as ours. I agree it will be better to use another name for our method, but we haven't come up with a better idea. Currently, we plan to continue using OrthoReg as our model name. Thanks for your suggestion!

---

> ### Comment · Reviewer_JMnM · 2022-11-17
> **Answer to Authors**
>
> Thank you for your responses! Except for the name of the method, they clarify all my concerns.

---

> > ### Author Response · Authors · 2022-11-17
> > **Response to Reviewer JMnM**
> >
> > Dear reviewer JMnM,
> >
> > Thanks again for your suggestions for improving this work. Considering that most of your concerns have been addressed, would you please consider raising the score?
> >
> > Best,
> > Paper 2282 authors

---

> > > ### Comment · Reviewer_JMnM · 2022-12-08
> > > **Response to authors**
> > >
> > > In light of other reviews and after discussion with the AC, I have decided to lower my score. The main reasons are:
> > >
> > > 1) The theoretical analysis is not complete (GzRn and Mv49)
> > > 2) The empirical analysis, which could alleviate this problem, does not show a clear collapse when using the supervised loss.

---

> > > > ### Author Response · Authors · 2022-12-09
> > > > **Response to the Reviewer**
> > > >
> > > > We thank the reviewer for providing more feedback, and we would like to clarify some points here.
> > > >
> > > > In the theoretical analysis in Sec.3, our intent is to study the effect of regularization terms on learned embeddings, so we didn't consider the effect of the supervised loss. Our analysis does show that the regularization term leads to the shrinking of the eigenspectrum of node embeddings, and such an effect may still exist even if the supervised loss is applied.
> > > >
> > > > We then verify it from the empirical side. Figure 2 and Figure 3 demonstrate that when the regularization loss is applied, the eigenspectrum tends to shrink as the training process proceeds (which can be justified by comparing the eigenvalue distribution when $\lambda = 0$ and $\lambda  = 0.001/0.1$). We understand that the reviewer might be concerned about the term 'collapse'. We use the concept of 'collapse' in this paper because similar phenomena (i.e., dimensional collapse) were previously studied in the context of self-supervised learning, where the supervised loss is never adopted. As a result, in those cases, the collapse phenomenon is 'complete collapse' (i.e., only the top-1 eigenvalue is preserved, and the remaining ones are almost negligible). In our cases, we can only observe the shrinking of the eigenspectrum of node embedding instead of complete collapse because of the effect of the supervised loss. Nevertheless, the empirical results in this section can verify that the regularization loss still has a significant impact on the embeddings' eigenspectrum in spite of the existence of the supervised loss.
> > > >
> > > > Anyway, we thank the reviewers' appreciation of the novelty and empirical contributions of our work. We wish to add more explanations to make the analysis in Sec.3 more clear and convincing if there is an opportunity.
> > > >
> > > > Best,
> > > >
> > > > Paper2282 Authors

---

### Official Review · Reviewer_GzRn · 2022-10-23

**Confidence:** 4
**Correctness:** 3
**Technical Novelty And Significance:** 2
**Empirical Novelty And Significance:** 3
**Recommendation:** 5

**Clarity, Quality, Novelty And Reproducibility:**

- The dimensionality collapse has been investigated in self-supervised learning, and the authors try to use it to find why GR-MLPs fail to match GNNs. Although the orthogonal penalty term is pretty common in recent works, the paper is well-motivated.

- The paper seems technically sound.

**Strength And Weaknesses:**

### Strength
- The paper is well-written and well-motivated. The idea and motivation are clarified and the paper is easy to follow.
- The experiments are strong and the performance could match the results of SOTA GNN models.

### Weakness
- The theoretical conclusions are based on the linearity assumption, which is not clarified in ABSTRACT. It is an important precondition of the theoretical analysis and it should be emphasized.
- When theoretically analyzing the existence of dimensionality collapse, the supervised loss, ($\ell_{xent}$, is neglected. It also produce the gradient w.r.t. W. However, Lemma 1 fails to clarify it. If I don't misunderstand, the conclusion only holds when $\lambda \rightarrow \infty$ ($\lambda$ is defined in Eq. 1). It is the main limitation of the anlaysis and the preconditions have to be clarified in Lemma 1.
- $\{\lambda_i^C\}$  seems to only appear in the beginning of theoretical analysis. I know the convergence of $\sigma_i(W(t))$ would lead to the statement, but there should be some descriptions of $\{\lambda_i^C\}_{i=1}$ after Lemma 1.
- As the authors claim that the efficiency of graph models is quite important, Eq. 5 seems a quite time-consuming operation. What is the difference between the GR-MLP with the proposed term and other models, including existing GR-MLPs and GNNs?

**Summary Of The Paper:**

This paper aims to investigate why the performance of graph-regularized methods could not match GNN models. It shows that the spectral collapse occurs under the linearity assumption of the mapping module (i.e., linear MLP). Then, the authors design a new regularization term for GR-MLPs. The experimental results show that the proposed term works pretty well on real datasets.

**Summary Of The Review:**

Overall, the paper is well-motivated and easy to follow. The theoretical analysis makes the orthogonal term convincing. The paper has some  merits, but there are several drawbacks as well. I'd like to update my score after reading the feedback and other reviews.

---

> ### Author Response · Authors · 2022-11-13
> **Responses to Reviewer GzRn**
>
> We thank the reviewer for the thorough reviews and kind suggestions. We note that the reviewer's concerns are mainly about the theoretical parts in Sec.3. E.g., some preconditions/assumptions are not emphasized and need more descriptions of the eigenvalues of the embedding matrix. Besides, there might be a misunderstanding of the claimed efficiency benefit of our method over GNNs. Here we'd like to address these concerns:
>
> Q1: Emphasize and clarify the assumptions for Lemma 1.
>
> A1: We thank the reviewer for pointing out that some preconditions and assumptions are not clearly stated. In the revised version, we have emphasized the linear model assumption in both the abstract and the introduction part of Lemma 1. However, we'd like to emphasize that a similar analysis can be extended to non-linear cases when the Universal Approximation Property of MLP is assumed. In Sec.3, we consider the linear model because it is much more convenient to analyze its gradient.
>
> We didn't consider the effect of supervised loss because in Lemma 1, we are mainly analyzing the effect of the regularization term on the gradient of the weight matrix. We agree that the supervised loss has a non-negligible effect on the gradient as well, but what we'd like to express through Lemma 1 is, the Laplacian regularization loss does have an effect to lead to dimensional collapse, even if the collapse phenomenon is not very severe. In the revised version, we have emphasized that we neglect the supervised loss and focus on the effect of Laplacian regularization in Lemma 1, and we hope in this way, the analysis in Sec.3 is clear and will not cause misunderstandings anymore.
>
> Besides, we'd like to point out that we clearly stated the two limitations: 1) linear model, 2) neglect of the effect of supervised loss, in Sec.3, the last paragraph before the empirical justifications. Furthermore, our empirical result in Sec.3 does not rely on the two assumptions, and it validates that the dimensional collapse phenomenon does exist.
>
>
> Q2: More descriptions of $\{\lambda_i^C\}_{i=1}^D$ after Lemma 1.
>
> A2: We have added more descriptions of $\{\lambda_i^C\}_{i=1}^D$ in Theorem 1 in the revised version. In theorem 1, we conclude that dimensional collapse behaves as the vanishing of small eigenvalues with respect to the largest few ones:
>
> $$
>    \lim_{t \rightarrow \infty} \frac{\lambda_i^{\mathbf{C}}(t)}{\lambda_j^{\mathbf{C}}(t)} = 0, \forall i \le d \; {\rm and} \; j \ge d+1.
> $$
> In this way, we can turn back to the analysis of the eigenvalues of embeddings $\mathbf{H}$ instead of $\mathbf{W}$. Besides, we also mentioned $\lambda_i^{\mathbf C}$ in the empirical justification part, where Fig. 2 plots the evolving of eigenvalues:
> $\{\lambda_i^C\}$, and Fig.3 plots the evolving of NESum
> ${\text{NESum}}(\mathbf{C}) = \{\lambda_i^{\mathbf{C}}\} \triangleq \sum_{i=1}^{d} {\lambda_i^{\mathbf{C}}} /{\lambda_1^{\mathbf{C}}}$.
>
> Q3: Differences between our method with existing methods with regard to efficiency.
>
> A3: We'd like to emphasize that the efficiency claimed in this paper is about the **inference stage** (instead of training). As stated in both Abstract and the first paragraph in Sec.1, GNNs rely on layer-wise message passing to aggregate features from the neighborhood, which is computationally inefficient during inference, especially when the model becomes deep and the graph is large. The training cost of GNNs and GR-MLPs (including ours) is similar, as all the methods require the utilization of the graph structure information, either explicitly or implicitly.
>
> The inference time, however, varies a lot. Consider a $L$-layer GNN, considering minibatch training (batch size is $B$) and neighbor sampling with a fixed number of neighbors as $k$, the inference complexity should be $O(BLd^2k^{L})$, while that for a $L$-layer MLP is only $O(BLd^2)$. As a result, MLPs (all GR-MLPs) can be inferred much faster than GNNs, especially when the graph is large and the model is deep. We also provide an empirical study in Fig.6, Appendix C.3 to show the superiority of our method over GNNs with regard to inference speed.

---

### Official Review · Reviewer_2EfD · 2022-10-23

**Confidence:** 4
**Clarity, Quality, Novelty And Reproducibility:** <covered in the above section>
**Correctness:** 3
**Technical Novelty And Significance:** 2
**Empirical Novelty And Significance:** 3
**Recommendation:** 6

**Strength And Weaknesses:**

Pros:
1. The authors take into account the related methods and explain their differences from their technique clearly. For example, differences from the graph-augmented MLPs were explained.
2. The idea of combining node embeddings and summary embeddings in Eq.7 is neat but with certain caveats, I believe. The choice of \alpha and \beta will play a vital role here.
3. The results in Table 1 are impressive
4. The paper in general clearly states the contribution and comparison with related works.

Comments
1. Extension to other graph types, Multi-graphs, graphs with hypernodes? As these types are typically used to represent knowledge graphs, do you foresee any issues with generalising the guarantees to these graph types?
2. (Page 3, 1st para) “Note that GNNs explicitly utilize the graph structure information through learning the mapping from node features and graph adjacency matrix to predicted labels. However, due to the above limitations, we seek for learning an MLP encoder, i.e., H = fθ(X) that only takes node features for making predictions.” Can you please explain the limitations? Are the limitations introduced by the graph structure, can you please give examples as GNNs perform quite well on graph structured data? (I might be missing some context here, in that case, kindly add more explanation in the draft. Maybe, here the authors might want to introduce the idea of soft-regularization.)
3. The theoretical results are for linear f(X). How well will they hold for the non-linear nature of MLPs? I only see empirical justification. This will be a strong paper if some theory can be developed in that direction, as pointed out in page 4.
4. What are the values of \alpha and \beta in table3, for the bottom 3 rows? I would like to understand how the regularization terms trade-off with each other. Thanks.
5. Have you studied other types of graph regularizations?


**Summary Of The Paper:**

This paper studies graph-regularized MLPs performance limitations. They show that the node embeddings space from a conventional GR-MLP suffers from dimensional collapse (or spectral collapse). Their solution, ORTHO-REG, mitigates this issue by introducing a soft regularization term based on the correlation matrix of node embeddings in the loss function and then removing the dependency of graph structure on the MLP loss term. Their experimental evaluation shows performance improvement over the SOTA.

**Summary Of The Review:**

Ablation study on the range of \alpha and \beta parameters done in section 5.4.2 highlights the strengths as well as limitations. Although, the authors demonstrate empirically that their regularization mitigates the spectral collapse problem, the theory proposed is not sufficient.

---

> ### Author Response · Authors · 2022-11-13
> **Responses to Reviewer 2EfD**
>
> We thank the reviewer for the valuable comments and positive feedback. We are glad to answer your questions.
>
> Q1: Extensions to other graph types, e.g., knowledge graphs.
>
> A1: Though this paper focuses on representation learning on homogeneous graphs, the analysis and the proposed method in this paper can naturally generalize to complex graphs like knowledge graphs. However, there may be some obstacles hindering the direct application of our method to these more complicated graph-structured data.
> 1) As we aim at using MLPs instead of GNNs for learning node embeddings, a very basic assumption should be that the node features are rich enough so that we can implicitly infer the structures using node features with our regularization technique. As The number of relation types of knowledge graphs can be large, the processed node features in knowledge graph datasets might be insufficient to infer both the existence and the type of an edge. A more promising way should be to combine the model with Langauge Models, e.g., we can generate node features using Language Models, which could be finetuned together with the top MLPs.
>
> 2) Different from homogeneous graphs, where there is only one type of node and edges, knowledge graphs usually have multiple types of nodes and edges. Note that to perform our regularization, we have to extract the neighborhood information with Eq.(6); this brings a challenge for knowledge graphs as we have to discriminate different types of neighborhoods according to the node types and edge types. One trivial way is to use different models for different edge types (like RGCN[1]), which means we require a regularization loss term for every edge type. This can be very inefficient. Besides, how to effectively fuse information from different edge types is also crucial and challenging.
>
> References:
>
> [1] Schlichtkrull, Michael, et al. "Modeling relational data with graph convolutional networks." European semantic web conference. Springer, Cham, 2018. https://arxiv.org/abs/1703.06103
>
>
> Q2: Explain the limitations of GNNs.
>
> A2: We are sorry for causing this ambiguous expression. Actually, in this sentence, we are mentioning the limitations of GNNs, which have been elaborated in Sec.1, the first paragraph:
>
> 1) GNNs rely on layer-wise message passing to aggregate features from the neighborhood, which is computationally inefficient during **inference**. For this limitation, you can refer to the analysis in GLNN[2] (Fig.1), which shows that the number of fetches and the inference time of GNNs are both magnitudes more than MLPs and grow exponentially as functions of the number of layers.
> 2) GNN models can not perform satisfactorily in cold-start scenarios where the connections of new incoming nodes are few or unknown. This could be observed if we study the classification accuracy of nodes with different degrees, like ColdBrew[3]. The conclusion is, the classification accuracy of nodes with a large degree is much higher than those of nodes with a low degree. Besides, GNNs perform poorly when predicting nodes having connections to the existing graph, even worse than the vanilla MLPs.
>
> References:
>
> [2] Zhang, Shichang, et al. "Graph-less neural networks: Teaching old mlps new tricks via distillation." In ICLR, 2022. https://arxiv.org/abs/2110.08727
>
> [3] Zheng, Wenqing, et al. "Cold Brew: Distilling graph node representations with incomplete or missing neighborhoods." In ICLR, 2022. https://arxiv.org/abs/2111.04840

---

> ### Author Response · Authors · 2022-11-13
> **Response to Reviewer 2EfD, Batch 2 of 2**
>
> Q3: Extend the analysis to non-linear cases.
>
> A3: Our theoretical analysis in Sec.3 is restricted to linear models as for a linear model, we are convenient to analyze the evolving of a single weight matrix. When extending to non-linear models, there will be more than one weight matrixes together with the effect of activation functions, which would be much more complicated.
>
> Despite this, we can still analyze the evolution of the final embeddings intuitively. The proof is based on the Universal Approximation Theorem[4], which shows that a shallow MLP model can approximate any function. With this assumption, we can treat node embeddings $\mathbf{H}$ as free learnable vectors. Then we can take the gradient of Laplacian regularization loss to $\mathbf{H}$:
> $$
> \frac{\partial \mathcal{L}_{reg}}{\partial \mathbf{H}} = \frac{\partial {{\rm tr}(\mathbf{H}^{\top}\mathbf{L}\mathbf{H})}}{\partial \mathbf{H}}  = \frac{\partial {{\rm tr}(\mathbf{H}^{\top}\mathbf{L}\mathbf{H})}}{\partial \mathbf{H}} \\\\
>  =  (\mathbf{L} + \mathbf{L}^{\top})\mathbf{H} = 2\mathbf{L}\mathbf{H} \\\\
> $$
> Similarly, we can treat the embedding matrix as a function of the training step $t$, i.e., $\mathbf{W} = \mathbf{H}(t)$, then we have $\frac{{\rm d}\mathbf{H}(t)}{{\rm d}t} = 2\mathbf{LH}$, and we can solve the equation analytically:
> $$
>     \mathbf{H}(t) = \exp(\mathbf{L}t) \cdot\mathbf{H}(0).
> $$
> The remaining proof is just similar to that in the proof of Lemma 1 in Appendix A.1.
>
> References:
> [4] Hornik, Kurt, Maxwell Stinchcombe, and Halbert White. "Multilayer feedforward networks are universal approximators." Neural networks 2.5 (1989): 359-366.
>
> Q4: What are the values of $\alpha$ and $\beta$ in Table 3, and what's the trade-off between them?
>
> A4: The $\alpha$ and $\beta$ in Table 3 are the same as in Table 1. Cora: $\alpha = 2e-3, \beta = 1e-6$; Citeseer: $\alpha = 1e-3, \beta = 1e-6$; Pubmed: $\alpha = 2e-6, \beta = 2e-6$. Although the optimal hyperparameters for different datasets are different, they are close in their values, and the model's performance is not very sensitive to their specific values. In the revised version, we add another section in Appendix C.5, studying how the trade-off between $\alpha$ and $\beta$. The conclusion is that when $\alpha / \beta$ is about $10^3$, our method can achieve satisfying performance as long as $\alpha$ is within a reasonable range, e.g., from $0.0005$ to $0.005$. With this observation, it will not be hard to deploy our method to a new dataset. For example, we can set the initial values as $\alpha = 0.001$, and $\beta = 10^{-6}$. Then we can slightly tune the two hyperparameters to find the optimal combination.
>
> Q5: Study other types of graph regularization.
>
> A5: Our analysis and experiments in Sec.3 study Laplacian regularization only but can be extended to other graph regularization methods that enforce smoothness of representations over the graph structure and without additional regularizations, e.g., Propagation Regularization (P-reg [5]), which is defined as follows:
> $$
> \mathcal{L}_{P-reg} =  \Vert \mathbf{\tilde{A}H} - \mathbf{H} \Vert_F^2.
> $$
>
> Similarly, we can take the gradient of $\mathcal{L}_{P-reg}$ to $\mathbf{H}$:
>
> $$
> \frac{\partial \mathcal{L}_{P-reg}}{\partial \mathbf{H}} = \frac{\partial  \Vert \mathbf{\tilde{A}H} - \mathbf{H} \Vert_F^2}{\partial \mathbf{H}} \\\\
>  =  2(\mathbf{\tilde{A} - I})^{\top}(\mathbf{\tilde{A} - I})\mathbf{H}\\\\
>  = 2 \mathbf{L}^{\top}\mathbf{L}\mathbf{H}.
> $$
> As a result, similar conclusions can be derived.

---

### Official Review · Reviewer_Mv49 · 2022-10-24

**Confidence:** 3
**Clarity, Quality, Novelty And Reproducibility:** On the whole, these are OK.
**Correctness:** 3
**Technical Novelty And Significance:** 3
**Empirical Novelty And Significance:** 3
**Recommendation:** 6

**Strength And Weaknesses:**

This paper demonstrates that the conventional GR-MLPs suffer from the dimensional collapse phenomenon in which a few large eigenvalues dominate the embedding space, thus restricting the representation power of these models through theoretical analysis and empirical results. To solve this problem, the authors propose ORTHO-REG, a GR-MLP model that adds regularization terms to encourage orthogonal embeddings and tackle high-order connectivity and non-homophily graphs. The model shows improved performance compared with conventional GR-MLPs and some GNNs on semi-supervised datasets and cold-start scenarios. In general, the paper explains the failure of existing GR-MLPs and is of good structure. However, there are some issues the authors need to address.

1. The theoretical analysis in Sec.3 is questionable. Lemma 1 and Theorem 1 only consider the laplacian smoothing term as the loss function and do not consider the effect of classification loss, which makes the whole theoretical analysis fundamentally problematic. In fact, the proof of lemma 1 implicitly assumes that the laplacian smoothing loss will be optimized to zero. However, in most cases, the smoothing term will not converge to zero because of the classification loss’s influence; thus, the relative value of the smaller eigenvalues to the largest one may not always decrease. Therefore, the lemma, as well as the theorem, does not hold.

2. In the experiment of Sec.3, the eigenspectra for node embeddings do not change much as the weight of the smoothing term increases, which may indicate the dimensional collapse phenomenon is not too severe. It also suggests that classification loss dominates the training process instead of smoothing loss, which contradicts the assumption of the above theoretical analysis.

3. The relation between the dimensional collapse phenomenon and the poor performance of existing GR-MLPs should be explained. Sec.3 mainly demonstrates the existence of the dimensional collapse phenomenon. However, there is no evidence that this phenomenon causes performance degradation of existing models. Intuitively, a modest dimensional collapse might help improve model performance because it might filter noise from input features.

4. Furthermore, the final model is inconsistent with the previous analysis because it introduces higher-order connectivity information, so it cannot be demonstrated that mitigating the dimensional collapse phenomenon alone will improve model performance. To verify this, experiments using the model based on eq.4 are suggested.


**Summary Of The Paper:**

This paper finds that the conventional Graph Regularized MLPs(GR-MLPs) suffer from the dimensional collapse phenomenon that a few large eigenvalues dominate the embedding space and proposes ORTHO-REG, a GR-MLP model that encourages orthogonal embeddings to mitigate the issue.

**Summary Of The Review:**

This paper tries to show that the dimensional collapse phenomenon leads to poor performance of the existing GR-MLP model, but there are defects in both the experiment and the theory, which makes a claim not strong enough.

---

> ### Author Response · Authors · 2022-11-13
> **Responses to Reviewer Mv49, Batch 1 of 2**
>
> We thank the reviewer for the valuable comments. We note that your concerns are mainly about the dimensional collapse phenomenon, i.e., whether it really exists and why collapsed representations are not desired. Here we would like to answer your questions one by one, and we hope this can address your concerns.
>
> Q1: Theoretical analysis in Sec.3
>
> A1: Our theoretical analysis here does focus on the Laplacian regularization term (we study the gradient of the regularization loss with respect to the weight matrix $\mathbf{W}$). This is reasonable as both the supervised cross-entropy loss function and Laplacian regularization are important for shaping the final learned node representations. Even if the supervised cross-entropy loss function can prevent completely collapsed representations, the gradient of the Laplacian regularization term evitably has such an effect (that makes different dimensions of node embeddings over-correlated). Besides, in the proof of Lemma 1, we didn't assume the smoothing term would be optimized to zero. The proof studies the evolving of the singular values of the weight matrix when optimized using the Laplacian regularization term with respect to the optimization step $t$. The conclusion is that the ratio between smaller eigenvalues and the largest one is monotonically decreasing as the training step $t$ increases, which can then lead to Lemma 1.
>
> We understand the reviewer's concern that in reality, the supervised loss function can alleviate this issue to some extent. However, we do point out that the theoretical analysis only works on the regularization term in the initially submitted version. Besides, we have clearly stated that the analysis has its limitations (see the last paragraph before the empirical justification in Sec.3). This is why we provide further empirical results (a non-linear model trained using a weighted sum of the supervised cross-entropy loss function and Laplacian regularization) to justify the existence of dimensional collapse phenomenon.
>
> Q2: Experiments of Sec.3 are not clear.
>
> A2: We are sorry for causing a misunderstanding of Fig.2. The reason that the differences of the eigenspectra as the weight term increases are difficult to discriminate is that we plot the changes of all $512$ eigenvalues. To better show the change. We replot Fig. 2, where we only focus on the decay of top-8 eigenvalues (we put the original figure to Fig.7 in Appendix C.4 as a reference). Besides, to better show the change of eigenspectra for increasing $\lambda$, we add another three subfigures that plot the top eigenvalues of three different $\lambda$ values at the same training epoch (epoch 0, 20, and 100). As suggested by Reviewer 4, we also plot the 95\% confidence intervals. In the updated Fig.2 (especially the lower three subfigures), we can easily observe that as $\lambda$ increases, the ratio of top eigenvalues (w.r.t. the largest one) decreases quicker. Specifically, at training epoch 100, the second largest eigenvalue for $\lambda = 0.001$ and $\lambda = 0.1$ becomes only $20\%$ of the largest one, indicating the largest eigenvalue contains most of the information, while the remainings are much less important. The empirical results are consistent with our analysis above that the dimensional collapse phenomenon does exist in Laplacian regularization, even if considering the supervised loss and a non-linear model.

---

> ### Author Response · Authors · 2022-11-13
> **Responses to Reviewer Mv49, Batch 2 of 2**
>
> Q3: Relationship between the dimensional collapse phenomenon and the poor performance of existing GR-MLPs.
>
> A3: We thank the reviewer for pointing out that the relationship between the dimensional collapse phenomenon of the poor performance of existing GR-MLPs is not clearly explained. Intuitively, when complete dimensional collapse happens, all data points will be embedded to fall on a line, and thus cannot be separated with a linear classifier. In Appendix E in the updated version, we further explain the limitation of weakly (or modestly) collapsed representations using a 2-dimensional example. Our main proposition is that, even if the training data points of modestly collapsed representations can be successfully separated by a linear classifier, they demonstrate worse robustness against attacks, and worse generalization ability to testing data (whose distribution might be a little bit shifted from the training one), due to the narrow embedding space on the direction of small eigenvalues.
>
> Another important thing is that when the dimensional collapse phenomenon exists, it can be hard to figure out whether it is a severe or moderate one. In this case, we'd rather directly eliminate the effect of dimensional collapse (as what we do in OrthoReg). Besides, enforcing orthogonality on node embeddings is just shaping the representations without regularization or constraint on the information that the representations should carry for downstream tasks. If the ability to filter noise is due to the supervised loss + lap-reg, enforcing orthogonality as what we do can hardly damage this ability as it uses the same embedding dimension, and it is just rebalancing the information each dimension should carry.
>
>
> Q4: Model inconsistency.
>
> A4: We do consider higher-order connectivity information in our final model, which is decided by the hyperparameter $T$. Note that when $T = 1$, our method only considers first-order connectivity, and we have studied the performance with different $T$ in Table 3 in the original version. When $T = 1$, our method can still achieve a very satisfying performance  (only a little bit lower than the best ones when $T = 2$). These results can validate that the improvement in the model's performance is not because of the high-order connectivity. Besides, these results also demonstrate that considering higher-order connectivity can bring improvement to the model, which was not considered in traditional GR-MLPs. This is also an important contribution of this work.

---

> ### Public Comment · ~Taiqiang_Wu1 · 2023-02-13
> **About GNN**
>
> The reviewer says that
> > Intuitively, a modest dimensional collapse might help improve model performance because it might filter noise from input features.
>
> Actually, GNN can be viewed as the feature filter with Laplace regularization $H^TLH$ [1]. So if the Laplace regularization leads to dimensional collapse, why do not the GNN meets the same problem ? I think more analysis are needed.
>
> Moreover, this paper follows the DirectCLR but DirectCLR belongs to unsupervised learning methods.
>
> [1] A Unified View on Graph Neural Networks as Graph Signal Denoising
>
> [2] Understanding Dimensional Collapse in Contrastive Self-supervised Learning

---

### Author Response · Authors · 2022-11-13
**General response to all reviewers**

We thank all the reviewers for their thorough comments and valuable opinions/suggestions. We've uploaded the revised version, which has addressed most of the concerns. Here we'd like to summarize the major changes in the revised version:

1) We emphasize the preconditions/assumptions of Lemma 1. Our Lemma 1 focuses on the Laplacian regularization loss, neglecting the impact of supervised loss. Besides, we assume a linear model to simplify the proof.

2) We replot Fig.2, focusing on the top-$8$ eigenvalues instead of all the $512$ eigenvalues, to better demonstrate the differences in different subfigures. Besides, we add another three subfigures in Fig.2, showing that at the same epoch, the drop rate of eigenvalues will increase as the trade-off hyperparameter $\lambda$ increases.

3) We add more content in Appendix: 1) In Appendix C.5, we conduct a sensitivity analysis of the trade-off hyperparameters $\alpha$ and $\beta$ in Eq.7. 2) in Appendix E, we provide more analysis of why dimensional collapse is not beneficial for linear classification.

4) We fix some typos and misleading symbols that might affect the reading.

Besides, we have provided individual responses for each reviewer, which answer each question detailedly. We hope our responses can address your concerns.

---

### Author Response · Authors · 2022-11-16
**We look forward to further responses from the reviewers**

Dear reviewers,

We provided detailed responses to address the reviewers' concerns last Saturday but haven't yet received further responses. We'd like to know whether you are satisfied with the responses and whether you have any other questions or concerns. We are willing to answer the follow-up questions if there are any.

Best,
Paper2282 Authors

---

### Decision · Program_Chairs · 2023-01-20

**Decision:**

Reject

**Justification For Why Not Higher Score:**

There are concerns with paper which requires a major revision and thereby another round of peer review.

**Justification For Why Not Lower Score:**

N/A

**Metareview: Summary, Strengths And Weaknesses:**

This paper investigates the dimensional collapse phenomenon, i.e., a few large eigenvalues dominate the embedding space, in conventional Graph Regularized MLPs (GR-MLPs). The paper then proposes ORTHO-REG, a GR-MLP model that encourages orthogonal embeddings to mitigate the issue. Both theoretical and empirical results are promising.

While the expert reviewers agree that the paper has some merits, there is a clear reservation as a result of the following major concerns:

- The theoretical analysis is incomplete (Reviewers GzRn and Mv49). Reviewer GzRn and Mv49 pointed out the issue with the theoretical analysis in Lemma 1, i.e., it neglects the supervised loss and the linearity assumption is made.  Lemma 1 fails to clarify why it is reasonable to consider only the Laplacian smoothing term without the supervised loss. Hence, it's unclear to what extent the theoretical result reflects what really happens in practice. After the rebuttal and discussion phases, the reviewers remain skeptical about this aspect of the paper.

- The empirical analysis, which could have alleviated this issue, does not show a clear collapse when using the supervised loss (Reviewer JMnM).

As a result, I cannot recommend this paper in its current form to be accepted for publication at ICLR 2023.

**Summary Of Ac-Reviewer Meeting:**

N/A